# Atypical cognitive training-induced learning and brain plasticity and their relation to insistence on sameness in children with autism

Jin Liu[1]*[†], Hyesang Chang[1][†], Daniel A Abrams[1], Julia Boram Kang[1], Lang Chen[1,2], Miriam Rosenberg-Lee[1,3], Vinod Menon[1,4,5]*

[1]Department of Psychiatry & Behavioral Sciences, Stanford University School of Medicine, Stanford, United States; [2]Department of Psychology, Santa Clara University, Santa Clara, United States; [3]Department of Psychology, Rutgers University, Newark, United States; [4]Department of Neurology & Neurological Sciences, Stanford Neurosciences Institute, Stanford, United States; [5]Stanford Neurosciences Institute, Stanford University School of Medicine, Stanford, United States

**\*For correspondence:**
jinliu5@stanford.edu (JL);
menon@stanford.edu (VM)

[†]These authors contributed equally to this work

**Competing interest:** The authors declare that no competing interests exist.

**Abstract** Children with autism spectrum disorders (ASDs) often display atypical learning styles; however, little is known regarding learning-related brain plasticity and its relation to clinical phenotypic features. Here, we investigate cognitive learning and neural plasticity using functional brain imaging and a novel numerical problem-solving training protocol. Children with ASD showed comparable learning relative to typically developing children but were less likely to shift from rule-based to memory-based strategy. While learning gains in typically developing children were associated with greater plasticity of neural representations in the medial temporal lobe and intraparietal sulcus, learning in children with ASD was associated with more stable neural representations. Crucially, the relation between learning and plasticity of neural representations was moderated by insistence on sameness, a core phenotypic feature of ASD. Our study uncovers atypical cognitive and neural mechanisms underlying learning in children with ASD, and informs pedagogical strategies for nurturing cognitive abilities in childhood autism.

## Editor's evaluation

This is an important study on learning strategy differences in autism vs typically developing controls. The study identifies similar learning rates but different learning strategies. The evidence provided by the authors is compelling, relying on well-done tasks and fMRI analyses. This paper will be of broad interest to the autism community and to the study of learning.

## Introduction

Autism spectrum disorder (ASD) is a neurodevelopmental disorder characterized by heterogenous profiles of cognitive functions (*Lord and Bishop, 2015*; *Lord et al., 2018*). While there have been reports of exceptional abilities in some domains such as calendrical calculation and veridical drawing in individuals with ASD (*Happé and Frith, 2010*; *Meilleur et al., 2015*; *Mottron et al., 2013*), prominent weaknesses in academically relevant cognitive domains, including math and reading, have also been reported in children and adolescents with ASD (*Jones et al., 2009*; *Mayes and Calhoun, 2003*; *Oswald et al., 2016*), likely arising from atypical learning styles. Relative to individuals with other

neurodevelopment disorders, many individuals with ASD achieve lower levels of post-secondary education, employment, and independent living (*Newman et al., 2011*; *Troyb et al., 2014*). Thus, there is a critical need for investigations of cognitive skill acquisition in ASD, and identifying the mechanisms of learning in affected children has taken on great significance and urgency.

Researchers have long been aware of the phenomenon of 'savant syndrome', in which some individuals with ASD demonstrate extraordinary skills in particular areas such as mathematics, music, and art (*Hughes et al., 2018*). Even among those without savant-level abilities, cognitive enhancements, beyond what is typically expected, have been observed in children with ASD (*Jones et al., 2018*; *Lord et al., 2018*). For example, certain types of problem-solving abilities appear to be enhanced in some children with ASD (*Uddin, 2022*). One plausible hypothesis is that alterations in learning styles and atypical neural pathways underlie enhanced cognitive functions in some individuals with ASD (*Chen et al., 2019*; *Iuculano et al., 2014*). However, the precise mechanisms underlying such potential enhancements, and sources of individual differences associated with the clinical phenotypic features of the disorder, remain unknown.

Although ASD is often conceptualized as a disorder of brain plasticity (*Church et al., 2015*; *Ecker and Murphy, 2014*), surprisingly, there have been few systematic investigations of the neurobiology of learning in children with ASD. An important consideration is that children with ASD often show a wide range of abilities, which are reflected in behavioral characteristics, cognitive abilities, and clinical features reported for this neurodevelopmental disorder (*Dinstein et al., 2012*; *Leekam et al., 2011*; *Lenroot and Yeung, 2013*; *Volkmar et al., 2004*). A high level of individual differences in cognitive abilities, even in high-functioning individuals with ASD, is now well documented with some individuals demonstrating remarkable abilities (*Baron-Cohen et al., 2007*; *Iuculano et al., 2020*; *Iuculano et al., 2014*; *Jones et al., 2009*; *Treffert, 2009*) and others showing marked deficits (*Bullen et al., 2020*; *Chen et al., 2019*; *Dowker, 2020*; *Oswald et al., 2016*). Despite accumulating evidence that suggests a wide range of cognitive abilities in ASD, little is known regarding whether individuals with ASD acquire cognitive skills in a similar or different way from their typically developing (TD) peers. Critically, to the best of our knowledge, it remains unknown whether distinct mechanisms of learning in children with ASD, following training, are reflected in plasticity of neural representations and whether there may be links between specific patterns of learning and brain plasticity and individual differences in clinical diagnostic features in affected individuals.

Previous work has highlighted plausible hypotheses regarding mechanisms of learning in children with ASD. One theoretical account suggests that the mechanisms of learning in children with ASD may be different from TD children (*Church et al., 2015*; *Gidley Larson and Mostofsky, 2008*; *Qian and Lipkin, 2011*). For example, it has been suggested that unlike their TD peers, children with ASD are biased toward memorizing specific examples, instead of learning complex regularities that enable generalization (*Qian and Lipkin, 2011*). Consistent with this view, individuals with ASD have demonstrated relative strengths in memorizing specific facts or associations (*Dawson et al., 2008*), but reduced ability to retrieve related items, compared to controls (*Cooper and Simons, 2019*). Other types of atypical mechanisms of learning in ASD have also been suggested, including hyper-systemizing characteristics that can lead to superior abilities for tasks that involve systematic and logical thinking and learning (*Baron-Cohen and Belmonte, 2005*; *Falter et al., 2008*). Although existing theories point to the possibility, there has been no direct evidence for atypical mechanisms of learning, at either the behavioral or neural level, in children with ASD in response to academically relevant interventions.

Here, we address these questions using a cognitive training program designed to improve numerical problem-solving skills, a domain that is critical for academic and professional success and achieving independence as an adult (*NMA Panel, 2008*; *Parsons and Bynner, 2005*; *Peters et al., 2006*; *Reyna and Brainerd, 2007*). Previous studies of numerical problem solving have shown that despite significant individual differences in performance (*Chen et al., 2019*), this cognitive domain represents a potential strength in children with ASD, with many showing preserved and even exceptional achievements (*Baron-Cohen et al., 2007*; *Iuculano et al., 2020*; *Iuculano et al., 2014*; *Jones et al., 2009*; *Treffert, 2009*). Crucially, little is known about individual differences in learning and brain plasticity in this domain in children with ASD, and, furthermore, their relation to phenotypic symptoms of restricted and repeated interests and behaviors (RRIB) associated with cognitive and behavioral inflexibility in ASD (*Crawley et al., 2020*; *Geurts et al., 2009*; *Uddin, 2021*).

We developed a theoretically motivated math training protocol combined with functional brain imaging acquired before and after training in 35 children with ASD (ages 8–11 years) and 28 age-, gender-, and IQ-matched TD children (*Figure 1*, *Figure 1—figure supplement 1*, *Supplementary file 1*, and *Supplementary file 2*). We had four goals. Our first goal was to investigate learning in response to math training in children with ASD compared to TD children. Our training protocol involved instructions on problem solving procedures and intensive practice on a set of math problems over five sessions, which has been previously shown to induce significant learning in TD children (*Chang et al., 2019*). Children's learning was assessed with multiple measures including trajectories of learning across five training sessions (learning profiles) and changes in performance between pre- and post-training sessions (learning gains). Crucially, we employed identical tasks before and after training and a standardized training protocol across the two groups. This approach enabled systemic analysis of learning in children with ASD relative to TD children. Based on previous reports of preserved or superior math performance in high-functioning individuals with ASD (*Baron-Cohen et al., 2007*; *Iuculano et al., 2020*; *Iuculano et al., 2014*; *Jones et al., 2009*; *Treffert, 2009*), we hypothesized that children with ASD would reveal comparable or even better learning, relative to TD children, for practiced, trained problems.

Our next goal was to examine whether learning gains were achieved through different cognitive mechanisms in children with ASD, when compared to TD children. This question would be particularly important to examine if learning outcomes following training were similar in the two groups. We quantitatively assessed individual differences in children's rule- and memory-based problem-solving strategy as a means of probing cognitive mechanisms of learning in children with ASD (*Iuculano et al., 2014*). In TD children, it has been shown that training focused on fluent problem solving leads to greater use of retrieval strategies for trained relative to novel, untrained problems (*Chang et al., 2019*), suggesting a shift from rule-based to memory-based learning in response to training. Consistent with previous accounts highlighting altered learning mechanisms in ASD (*Church et al., 2015*; *Gidley Larson and Mostofsky, 2008*; *Qian and Lipkin, 2011*), we hypothesized that children with ASD would show distinct mechanisms of learning compared to TD children, reflected by different patterns of shift in problem-solving strategy in response to training.

The third goal of our study was to investigate the neural mechanisms of learning in children with ASD compared to TD children. Functional brain imaging studies in TD children have revealed the involvement of distributed brain areas involved in math problem solving and learning (*Butterworth and Walsh, 2011*; *Menon, 2016*; *Menon and Chang, 2021*; *Piazza and Eger, 2016*). Structures of the medial temporal lobe (MTL), including the hippocampus and parahippocampal gyrus, are particularly important for acquiring numerical problem-solving skills in children, consistent with their role in learning and memory (*Menon, 2016*; *Menon and Chang, 2021*). Additionally, the intraparietal sulcus (IPS), which is crucial for representing and manipulating quantities (*Butterworth and Walsh, 2011*; *Piazza and Eger, 2016*), is thought to play an important role in math learning. For example, previous work has shown that short-term math training induces plasticity of functional brain circuits linking the MTL and IPS, which is associated with individual differences in learning or changes in memory-retrieval strategy use during math problem solving in TD children (*Jolles et al., 2016*; *Rosenberg-Lee et al., 2018*). Importantly, it remains unknown whether children with ASD exhibit altered patterns of changes in functional brain systems associated with math learning, compared to TD children. Based on a cross-sectional study that observed heterogeneous profiles of brain activation in relation to numerical problem solving in ASD (*Iuculano et al., 2020*), we reasoned that atypical mechanisms of mathematical learning in children with ASD would result in distinct patterns of learning-related neural representational plasticity in the MTL and IPS.

The fourth and final goal of our study was to investigate the influence of a core autism symptom domain, RRIB (*Bishop et al., 2006*; *Faja and Nelson Darling, 2019*; *Kanner, 1943*; *Qian and Lipkin, 2011*), on the relation between brain plasticity and learning in children with ASD. It has been proposed that RRIB in ASD may be associated with over-reliance on regularities and rules (*Baron-Cohen and Lombardo, 2017*; *Baron-Cohen et al., 2003*). Although behavioral studies have proposed a connection between RRIB and flexible behavior during probabilistic reversal learning tasks (*Crawley et al., 2020*; *D'Cruz et al., 2013*), the neural mechanisms linking RRIB to learning have not been explored. To determine whether RRIB contributes to atypical mechanisms of learning in children with ASD, we evaluated the hypothesis that RRIB symptoms would influence the relation between brain plasticity

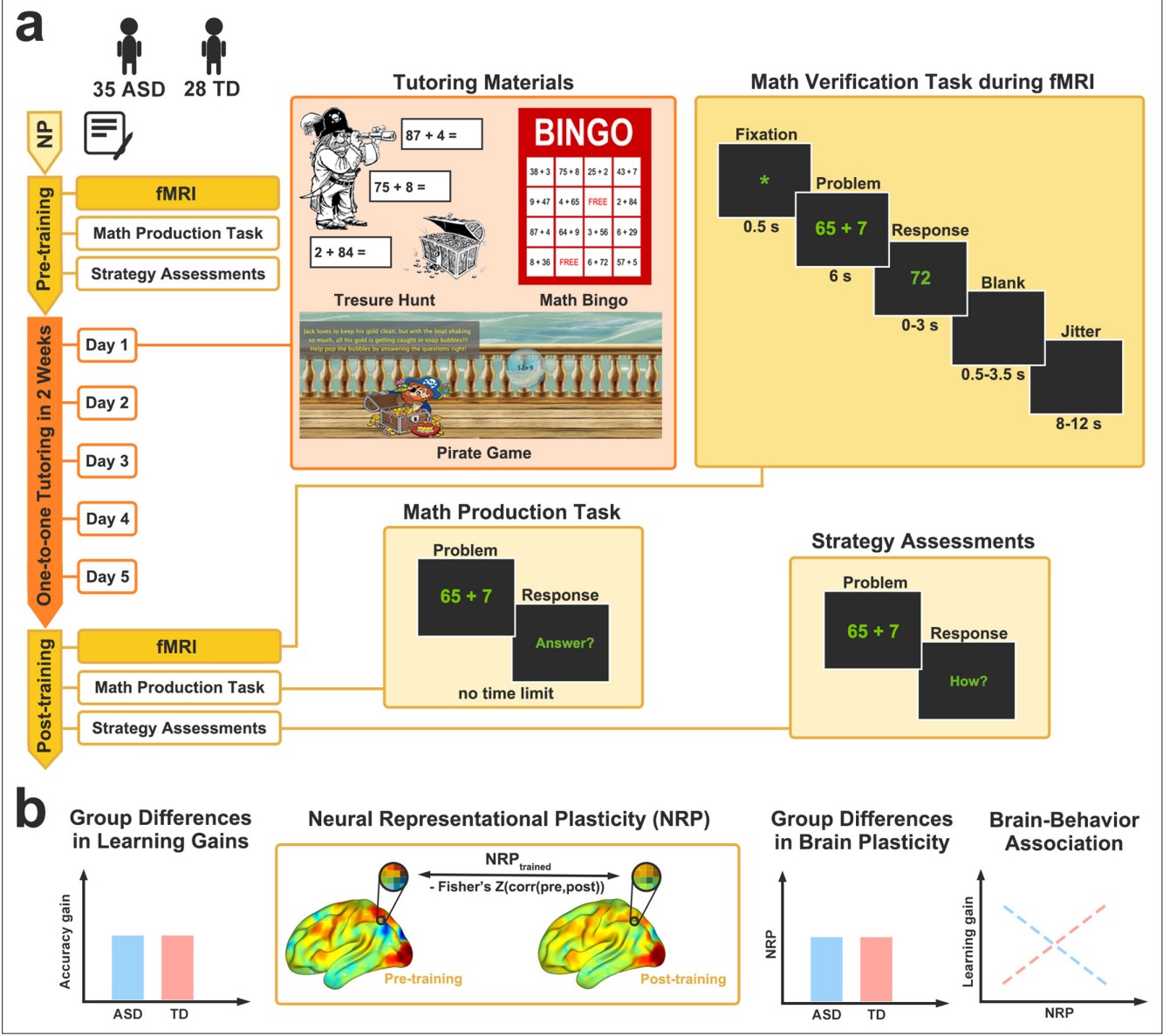

**Figure 1.** Overview of study design and analysis approach. (**a**) *Study design*. Before training, all eligible children with autism spectrum disorder (ASD) and typically developing (TD) children underwent neuropsychological (NP) assessments, a functional magnetic resonance imaging (fMRI) scan session (math verification task), and tasks outside of the scanner (math production task and strategy assessment). Children completed 5 days of one-to-one math training. On each training day, children completed multiple interactive activities with a tutor, including *Treasure Hunt*, *Math Bingo*, and *Pirate Game* (Materials and methods). After training, children completed a second fMRI scan session and outside-of-scanner tasks. In the math verification task, each trial began with a 500ms fixation cross, followed by a 6 s presentation of a math problem. Participants considered the problem and then indicated if a subsequent solution probe matched their answer. In the math production task, children were required to solve each addition problem by verbalizing their answers. After solving each problem, children's problem-solving strategies were assessed, in which they were asked to describe how they solved the problem. Each task fMRI run lasted 4 min and 50 s. Children's performance on trained and untrained (novel, similar-to-trained) problems was assessed by a math verification task during fMRI and a math production task before and after training. Children's problem-solving strategy use (memory-based or rule-based strategy) for trained and untrained problems was assessed using strategy assessments (Materials and methods). (**b**) *Analysis approach*. We first assessed learning, using multiple measures including learning gain, in children with ASD and TD children and then investigated whether cognitive and neural mechanisms of learning are altered in children with ASD, compared to TD children. Training-related brain plasticity was assessed by neural representational plasticity (NRP), multivariate spatial correlation of brain activation patterns between pre- and post-training for trained problems. Brain imaging analyses included comparisons between ASD and TD groups, using both whole-brain and region of interest analysis approaches. Schematic graphs illustrate possible outcomes.

The online version of this article includes the following figure supplement(s) for figure 1:

**Figure supplement 1.** Participant inclusion and exclusion procedures.

*Figure 1 continued on next page*

*Figure 1 continued*

**Figure supplement 2.** Problem sets used in two tranining problem sets, Set A and Set B.

and learning in these children. We were particularly interested in the contribution of insistence on sameness (IS), a core phenotypic feature of ASD related to cognitive and behavioral inflexibility and resistance to changes in routine (*Lam et al., 2008*; *Supekar et al., 2021b*). We hypothesized that, among the three RRIB elements (*Supekar et al., 2021b*), IS would be most closely associated with atypical learning patterns, as cognitive flexibility – which can vary from highly flexible and adaptive to less flexible and adaptive – has the potential to significantly shape an individual's learning style.

## Results

### Learning profiles of children with ASD, compared to TD children, during training

The first goal of the study was to examine whether children with ASD demonstrate comparable learning, relative to TD children. To address this, we first examined children's learning profiles across 5 days of math training (*Figure 1a*). On each day of training, performance on trained math problems was assessed using an inverse efficiency score (*Bruyer and Brysbaert, 2011*), measured as problem reaction time divided by accuracy. Higher scores on this measure indicated poorer performance. A 5x2 (Session x Group) repeated measures ANOVA showed a significant main effect of session ($F$(4, 212)=41.82, p<0.001, $\eta^2_p$=0.44, BF>100), but no main effect of group, or session by group interaction ($F$s≤0.09, ps≥0.471, BFs<0.33). A one-way repeated measures ANOVA indicated that both children with ASD and TD children showed significant changes in inverse efficiency scores across five days of training ($F$s≥18.63, ps<0.001, $\eta^2_p$=0.43, BFs≥100) (*Figure 2a* and *Supplementary file 3*). Analysis of accuracy and reaction time revealed similar improvements across groups. For accuracy, a 5x2 (Session x Group) repeated measures ANOVA showed a significant main effect of session ($F$(4, 212)=8.21, p<0.001, $\eta^2_p$=0.13), but no main effect of group or session by group interaction ($F$s≤1.01, ps≥0.405). For reaction time, a 5x2 (Session x Group) repeated measures ANOVA showed a significant main effect of session ($F$(4, 212)=127.81, p<0.001, $\eta^2_p$=0.71), but no main effect of group or session by group interaction ($F$s≤1.42, ps≥0.230). These findings indicate that children with ASD improved as well as TD children across training sessions, on both accuracy and reaction time for trained math problems.

Additionally, children's learning rate across five training days, derived from a linear regression model, was comparable between children with ASD and TD children ($t$(53)=–0.86, p=0.391, BF=0.27). To further examine group differences in performance across training days, planned two-sample $t$-tests were performed for each training day. This analysis confirmed that inverse efficiency scores were not significantly different between groups on any training day (|$t$s|≤0.84, ps≥0.41, BFs<0.46) (*Supplementary file 4*). These results indicate that across five days of math training, children with ASD and TD children showed comparable learning profiles.

### Learning gains in children with ASD, compared to TD children, in response to training

Next, we examined children's training-related learning gains on two math problem-solving tasks, verification and production tasks (*Figure 1a*). Children's performance was assessed by accuracy in the verification task and reaction time in the production task (see **Materials and methods** for details). In these two tasks, both trained and untrained (novel problems similar to trained) problems were presented before and after training. Our analysis focused on assessing training-related learning gains in children's performance on trained problems. To address the specificity of learning gains for trained problems, we additionally examined children's performance gains on untrained problems.

#### Verification task

A 2x2 (Time x Group) repeated measures ANOVA showed a significant main effect of time ($F$(1,61)=9.09, p=0.004, $\eta^2_p$=0.13, BF=7.75), and no significant main effect of group or group by time interaction ($F$s≤1.60, ps≥0.210, BFs<0.59) on accuracy for trained problems (*Figure 2b*, **top** and *Supplementary file 3*), suggesting significant learning gains across the two groups. Post-hoc paired t-tests revealed

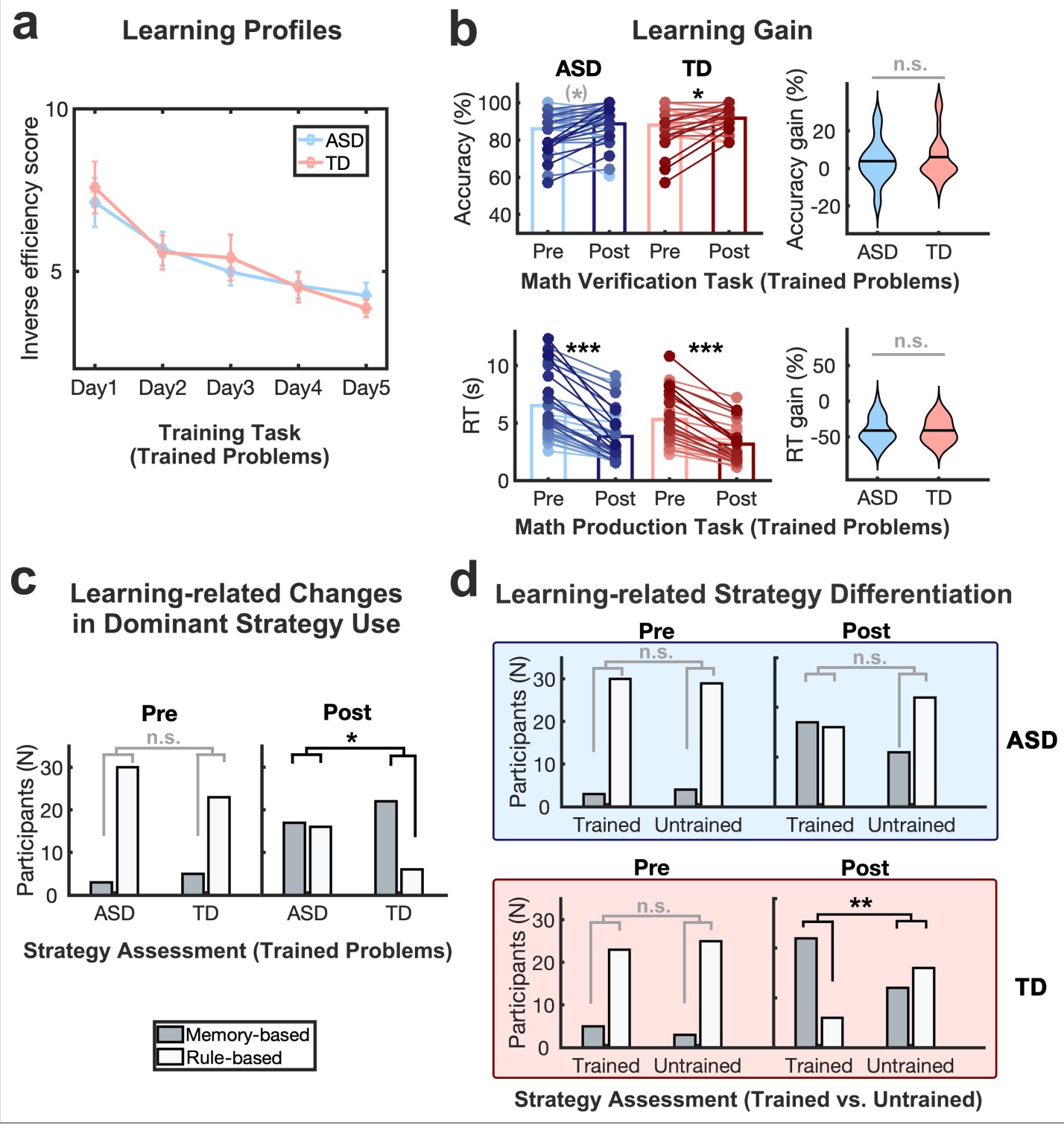

**Figure 2.** Similar learning outcomes but altered cognitive mechanisms of learning in response to training in children with autism relative to typically developing children. (**a**) *Learning profiles*. Children with autism spectrum disorder (ASD) showed significant improvements in inverse efficiency score (reaction time/accuracy) of the training task from training day 1–5 at a comparable level to typically developing (TD) children. Error bars represent standard errors of the mean. Sample size: n(ASD)=29 and n(TD)=26. (**b**) *Learning gains*. Training improved performance on trained problems in children with ASD at comparable levels to TD children, with no significant differences in learning gains between groups in accuracy in math verification task or reaction time (RT) in the math production task. The pair of dots connected by a line represents the performance of the same child at pre- and post-training, with darker blue/red dots/lines indicating greater performance gains. Group means are presented in lighter and darker blue/red bars at pre-

*Figure 2 continued on next page*

*Figure 2 continued*

and post-training. Sample size for math verification task: n(ASD)=35 and n(TD)=28; Sample size for math production task: n(ASD)=33 and n(TD)=28.
(c) *Changes in dominant strategy use for trained problems.* At pre-training, children with ASD and TD children showed comparable distribution of dominant strategy use (memory-based or rule-based) for trained problems. At post-training, the distribution of dominant strategy use was significantly different between groups for trained problems. While most TD children used the memory-based strategy most frequently following training, nearly half of the children with ASD used rule-based strategies most frequently for trained problems. (d) *Differentiation of dominant strategy use between trained and untrained problems.* No difference in the distribution of dominant strategy use between types of problems was observed at pre-training. After training, the distribution of strategy use was not significantly distinguishable between trained and untrained problems in children with ASD, whereas TD children reported significantly greater use of memory-retrieval strategy for trained problems compared to untrained problems. Sample size: n(ASD)=33 and n(TD)=28. (*) p<0.1; * p<0.05; ** p<0.01; *** p<0.001; *n.s.,* not significant.

The online version of this article includes the following figure supplement(s) for figure 2:

**Figure supplement 1.** Changes in performance and strategy use for untrained problems in children with autism spectrum disorder (ASD) and typically developing (TD) children in response to training.

greater accuracy for trained problems at post-training relative to pre-training in TD children ($t(27)=-2.54$, p=0.017, Cohen's $d=-0.48$, BF=2.91) and a marginally significant effect of time in children with ASD ($t(34)=-1.72$, p=0.095, Cohen's $d=-0.29$, BF=0.68). The difference in accuracy gain between groups was not significant ($t(61)=-0.73$, p=0.470, BF=0.32). For untrained problems, there were no significant main effects of group or time or group by time interaction ($Fs\leq1.90$, $ps\geq0.173$, BFs<0.55; *Figure 2—figure supplement 1* and *Supplementary file 3*, *Supplementary file 4* ).

## Production task

A 2x2 (Time x Group) repeated measures ANOVA using reaction time revealed a main effect of time ($F(1,59) = 148.13$, p<0.001, $\eta^2_p=0.72$, BF>100), and no significant main effect of group or group by time interaction ($Fs \leq2.87$, $ps \geq0.096$, BFs<0.87) for trained problems (*Figure 2b,* **bottom** and *Supplementary file 3*). Post-hoc paired t-tests revealed a significant decrease in reaction time for trained problems for both TD and ASD groups ($ts\geq8.68$, ps<0.001, Cohen's $d\geq1.51$, BFs>100) (*Supplementary file 3*), and this reaction time decrease (learning gain) was comparable between groups ($t(59)=-0.03$, p=0.979, BF=0.28). For untrained problems, a 2x2 (Time x Group) repeated measures ANOVA using reaction time revealed a significant group by time interaction ($F(1,59)=6.06$, p=0.017, $\eta^2_p=0.09$, BF=4.28), in which the ASD group showed a significant reduction in reaction time between pre- and post-training ($t(32)=3.70$, p<0.001, Cohen's $d=0.64$, BF=38.68), while the TD group did not ($t(27)=1.35$, p=0.187, BF=0.46) (*Figure 2—figure supplement 1* and *Supplementary file 3*, *Supplementary file 4* ).

Taken together, results from verification and production tasks provide converging evidence that learning gains on trained problems in response to five days of math training were comparable between children with ASD and TD children. Moreover, children's learning gains were specific to trained problems in the TD group, while children with ASD demonstrated reduced reaction times not only on trained problems but also on untrained problems, which provides some evidence for positive transfer in ASD.

## Changes in problem-solving strategy use in response to training in children with ASD, compared to TD children

The second goal of the study was to examine cognitive mechanisms of learning in response to training in children with ASD, compared to TD children, with a focus on changes in math problem-solving strategies using well-validated strategy assessments (*Wu et al., 2008*). Children's dominant strategy was determined by the most frequently reported strategy across correctly solved problems in each trained and untrained condition at each time point. To examine whether training leads to differential use of rule-based (counting, decomposition) or memory-based (memory-retrieval) strategies in children with ASD relative to TD children, we first compared children's strategy use for trained problems before and after training between the two groups. Prior to training, children with ASD and TD children showed comparable strategy use for trained problems ($\chi^2_1=0.04$, p=0.312, BF=0.50) (*Supplementary file 5*). However, group differences were evident following training. Specifically, while most TD children used the memory-based strategy most frequently following training, nearly half of the children with ASD used rule-based strategies most frequently for trained

problems ($\chi_1^2$=4.81, p=0.028, $\phi$=0.28, BF=3.39) (**Figure 2c**). For untrained problems, children with ASD and TD children showed comparable strategy use both before and after training ($\chi^2 \leq 0.58$, ps≥0.444, BFs<0.46; **Figure 2—figure supplement 1** and **Supplementary file 5**), confirming the specificity of group differences in problem-solving strategy use for trained problems in response to training.

To further determine whether training leads to greater changes in strategy use for trained relative to untrained problems following training, we next examined children's reliance on rule- and memory-based strategies for trained and untrained problems before and after training in each group. Results showed that before training, the distribution of rule- and memory-based strategy use in children with ASD and TD children was comparable across trained and untrained problems ($\chi^2 s \leq 0.58$, ps≥0.445, BFs<0.57; **Supplementary file 5**). In contrast, following training, TD children showed significantly greater use of memory-based strategies for trained relative to untrained problems ($\chi_1^2$=7.49, p=0.006, $\phi$=0.37, BF=13.24), while children with ASD continued to use similar problem-solving strategies between trained and untrained problems ($\chi_1^2$=2.23, p=0.135, BF=0.90; **Figure 2d**). Additionally, to estimate how consistently the dominant strategy is used in each participant, we calculated the dominant strategy rate by dividing the number of problems using the dominant strategy by all possible problems (i.e. 14 trained problems). No significant group differences on dominant strategy rate were found either before (two-sample t-test, t(59)=0.41, p=0.683; ASD: M=0.67, SD=0.21; TD: M=0.65, SD=0.18) or after training (two-sample t-test, t(59)=−0.48, p=0.630; ASD: M=0.76, SD=0.21; TD: M=0.78, SD=0.20).

These results demonstrate that despite similar improvements in post-training performance between children with ASD and TD children, the problem-solving strategies employed by these two groups diverged significantly. Specifically, even with a training protocol designed to encourage memory-based problem-solving strategies, children with ASD showed a greater tendency to rely on rule-based learning for trained problems, as opposed to the memory-based learning more frequently employed

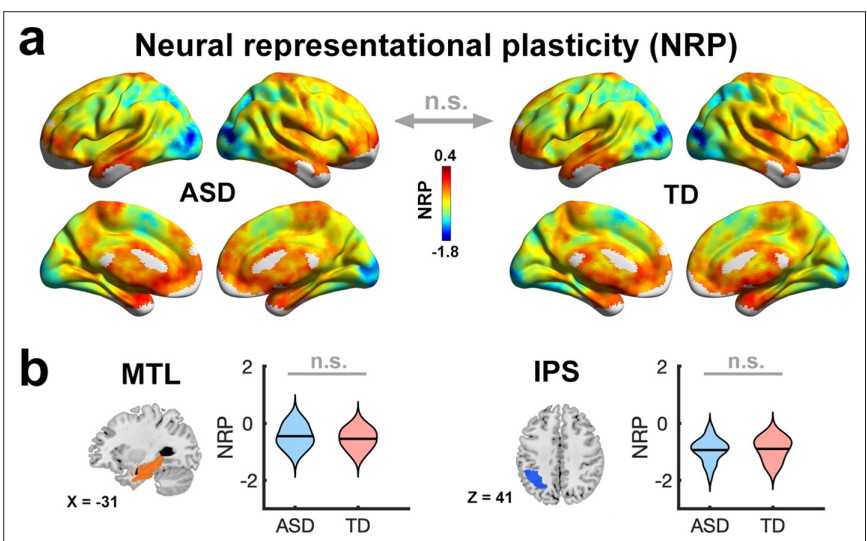

**Figure 3.** Comparable training-related neural representational plasticity (NRP) in children with autism and typically developing children. Both (**a**) whole-brain and (**b**) region of interest (ROI) multivariate neural representational pattern analysis showed no significant differences in mean NRP across individuals between children with autism spectrum disorder (ASD) and typically developing (TD) children. The results of left hemisphere ROIs are shown here, and the results of right hemisphere ROIs are shown in **Figure 3—figure supplement 1**. Sample size: n(ASD)=21 and n(TD)=24. MTL, medial temporal lobe; IPS, intraparietal sulcus; *n.s.*, not significant.

The online version of this article includes the following figure supplement(s) for figure 3:

**Figure supplement 1.** Group differences in neural representation plasticity (NRP) for right hemisphere regions of interest (ROIs).

by TD children. Additionally, children with ASD exhibited less variation in strategy use between trained and untrained problems in response to training, in comparison to their TD peers.

## Training-related neural representational plasticity in children with ASD, compared to TD children

The third goal of the study was to examine whether children with ASD reveal distinct neural mechanisms of learning relative to TD children. Extending beyond canonical univariate analysis methods, we used multivariate neural representational pattern analysis (*Iuculano et al., 2015*; *Kragel et al., 2018*; *Kriegeskorte et al., 2008*) to characterize functional brain plasticity in response to training on a fine spatial scale. We examined *neural representational plasticity (NRP)* using a brain-based distance measure that computes differences in brain activity pattern during math problem solving between pre- and post-training (*Figure 1b*; see also **Materials and methods**). Higher NRP scores indicated relatively greater training-related brain plasticity.

### Group differences

We first examined whether children with ASD and TD children show group differences for NRP. Our whole brain analysis showed that the two groups are similar for this measure (*Figure 3a*). To further examine the plasticity of key brain regions associated with numerical problem solving and learning (*Butterworth and Walsh, 2011*; *Menon and Chang, 2021*), we performed the analysis using a priori defined MTL and IPS regions (see **Materials and methods**). We found that NRP was comparable between groups in these regions (|*t*s|≤0.81, ps≥0.42; left: *Figure 3b* and right: *Figure 3—figure supplement 1*).

### Relation between NRP and learning gains

We then examined whether ASD and TD groups show similar or different relationships between NRP and individual differences in learning using a general linear model (independent variables: group, learning gains, and their interaction; dependent variable: NRP). In this analysis, learning gains were computed as changes in accuracy for trained problems in the verification task during the fMRI scan. Results from whole brain analysis revealed a significant group by learning gain interaction in multiple distributed regions, including those typically associated with math learning. Specifically, TD children showed a positive relationship between NRP and learning gains, indicating that greater brain *plasticity* is associated with improvements in performance following training in the TD group. In contrast, children with ASD revealed a negative relationship between NRP and learning gains, indicating that greater neural *stability* is associated with improvements in performance following training in children with ASD. These dissociable brain-behavior relationships were evident in the bilateral MTL, right IPS, right lateral occipital cortex, right frontal eye field, and right middle frontal gyrus (*Figure 4* and *Supplementary file 6*).

Results using a priori defined regions revealed similar brain-behavior relationships in bilateral MTL and left IPS regions as observed in the whole brain analysis (*Figure 4—figure supplement 1* and *Supplementary file 7*). Additional analysis revealed that these findings are specific for trained problems: no significant interaction between group and changes in performance was observed for untrained problems in these brain regions (*Figure 4—figure supplements 2–3*).

Together, findings indicate that while children with ASD and TD children demonstrate comparable extent of functional brain plasticity following training, learning is supported by greater neural plasticity in the MTL, IPS, lateral occipital, and frontal regions in TD children, and, in contrast, by greater neural stability in these brain systems in children with ASD.

## Influence of insistence on sameness on the relationship between training-related brain plasticity and learning in children with ASD

The final goal of the analysis was to investigate the role of clinical symptoms (RRIB), and specifically insistence on sameness (IS), in atypical relations between brain plasticity and learning in children with ASD. We performed a moderation analysis (*Figure 5a*) to examine how IS influences the relationship between learning and NRP following training. This analysis focused on the MTL and IPS, brain regions

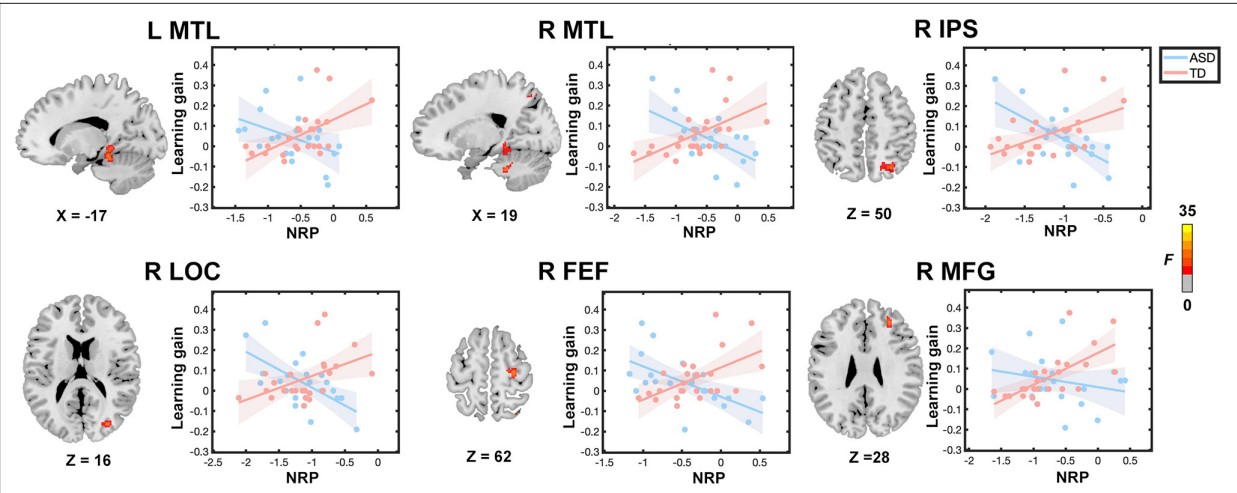

**Figure 4.** Atypical relation between training-related neural representational plasticity (NRP) and learning gains in children with autism. Whole-brain multivariate neural representational pattern analysis revealed a significant group by learning gain interaction in the bilateral medial temporal lobe (MTL), right intraparietal sulcus (IPS), right lateral occipital cortex (LOC), right frontal eye field (FEF), and right middle frontal gyrus (MFG). NRP scores were normalized values of correlation between pre- and post-training multiplied by –1. Larger NRP scores indicated a greater degree of training-related brain plasticity, while smaller NRP scores indicated a lower degree of plasticity in neural representation between pre- and post-training, reflecting increased stability of neural representations over time. In typically developing (TD) children, shown in the pink line, greater learning gains were associated with greater NRP. In children with autism spectrum disorder (ASD), shown in the blue line, greater learning gains were associated with lower NRP, indicative of more stable neural representations. NRP in each region identified from the interaction between group and learning gains in whole-brain analysis was extracted from 6 mm spheres centered at peaks for visualization of results. Shaded areas indicate 95% confidence intervals. Sample size: n(ASD)=21 and n(TD)=24.

The online version of this article includes the following figure supplement(s) for figure 4:

**Figure supplement 1.** Results of brain-behavior association between region of interest (ROI)-based neural representational plasticity (NRP) and learning gains for trained problems in children with autism spectrum disorder (ASD), compared to typically developing (TD) children.

**Figure supplement 2.** Brain-behavior association between neural representational plasticity (NRP) and changes in performance for untrained problems in children with autism spectrum disorder (ASD) relative to typically developing (TD) children.

**Figure supplement 3.** Brain-behavior association between region of interest (ROI)-based neural representational plasticity (NRP) and changes in performance for untrained problems in children with autism spectrum disorder (ASD) relative to typically developing (TD) children.

**Figure supplement 4.** Correlation between neural representational plasticity (NRP) and rule-based strategy persistence in children with autism spectrum disorder (ASD).

strongly implicated in numerical problem solving and learning (**Butterworth and Walsh, 2011**; **Menon and Chang, 2021**) and identified as significant predictors of learning in this study.

Here, we found that IS moderates the relation between NRP and learning gains in children with ASD in the left MTL ($b=-0.85$, se=0.31, $t=-2.76$, p=0.015) and right IPS ($b=-0.44$, se=0.19, $t=-2.28$, p=0.038; **Figure 5b–c** and **Supplementary file 8**). Specifically, the relationship between reduced NRP and greater learning gains in children with ASD was driven by individuals with higher levels of IS. To address whether the findings are specific to IS, we additionally examined two other RRIB components (circumscribed interests and repetitive motor behaviors). This analysis showed that the other two RRIB components were not significant moderators of NRP - learning relation ($|ts|\leq1.28$, ps≥0.091). To address whether these findings are specific to MTL and IPS regions, we examined whether IS moderates the relation between NRP of other brain regions and learning gains. We found no significant moderation effect for either model that included NRP of a visual region (V1; $b=-0.22$, se=0.25, $t=-0.86$, p=0.406) or the whole brain ($b=-0.29$, se=0.32, $t=-0.93$, p=0.370) in children with ASD. Together, these results suggest that cognitive features of clinical RRIB symptoms - insistence on sameness - contribute to atypical mechanisms of learning in children with ASD.

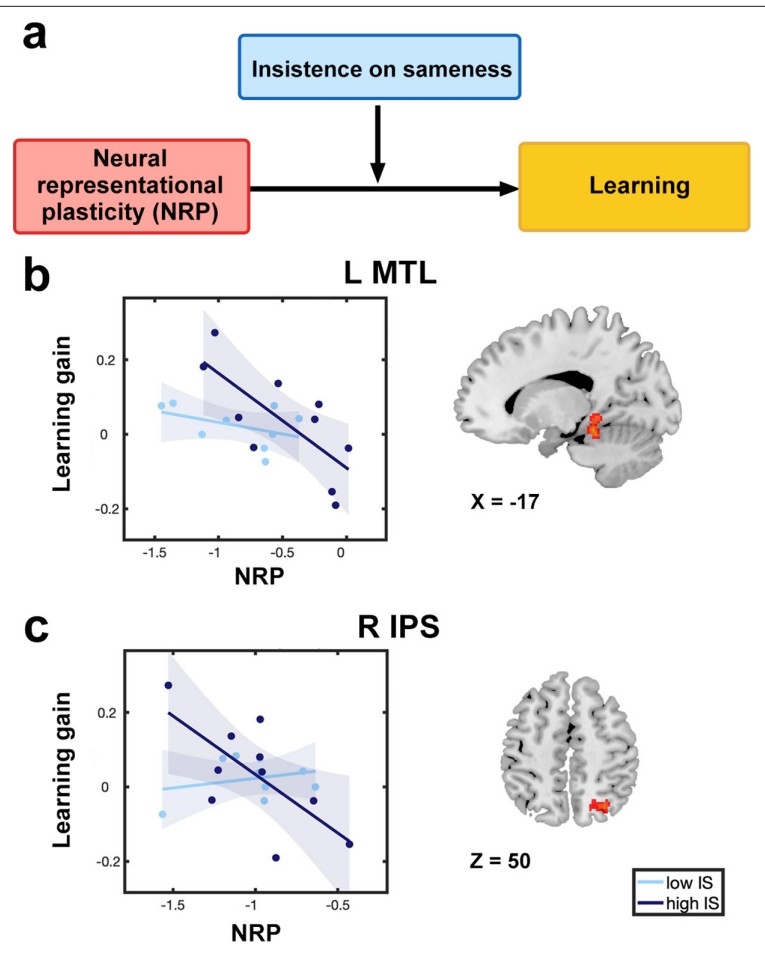

**Figure 5.** Insistence on sameness moderates the relation between training-induced brain plasticity and learning in children with autism. (**a**) A moderation analysis was performed to examine whether insistence on sameness (IS), a cognitive component of clinical symptoms (RRIB) in autism, influences the relation between functional brain plasticity and learning. (**b–c**). IS moderates the relation between neural representational plasticity (NRP) and learning gains in (**b**) the left medial temporal lobe (MTL) and (**c**) right intraparietal sulcus (IPS). Higher levels of IS were linked to a more atypical association between NRP and learning gains in children with autism spectrum disorder (ASD). The sample of ASD participants was median-split into high and low IS groups based on the scores on IS to illustrate atypical brain-behavior association moderated by the severity of IS symptoms. Shaded areas indicate 95% confidence intervals. Sample size: n(ASD)=19.

The online version of this article includes the following figure supplement(s) for figure 5:

**Figure supplement 1.** Strategy transition moderates atypical relation between training-induced brain plasticity and learning in children with autism spectrum disorder (ASD).

## Discussion

We examined whether cognitive and neural mechanisms of learning are altered in children with ASD, compared to TD children, using a theoretically motivated training protocol and multivariate neuro-imaging analysis. We found that despite comparable learning between children with ASD and TD children in response to training in numerical problem solving, children with ASD relied on different cognitive mechanisms to support mathematical problem-solving skill acquisition, compared to TD peers. Our multivariate neural representational pattern analysis revealed functional brain plasticity following training in both children with ASD and TD children. Critically, although overall changes in brain and behavior were similar between groups, we observed divergent relationships between learning and plasticity in brain systems typically associated with math learning: while TD children showed a positive relationship between plasticity and learning gains, children with ASD revealed a negative relationship between plasticity and learning gains, indicating that more stable neural

representations are associated with performance improvements in these children. Lastly, we found that a hallmark of ASD symptomology, insistence on sameness, contributes to atypical associations between brain plasticity and learning in children with ASD. Our study reveals atypical cognitive and neural mechanisms of learning in childhood autism, and informs effective pedagogical interventions to remediate and nurture cognitive abilities in affected individuals.

## Learning following cognitive training in children with ASD

The first major finding of this study is that learning, computed across multiple behavioral measures, is comparable between children with ASD and TD children following an academically relevant training program. These results suggest that high-functioning children with autism have a preserved ability to acquire cognitive skills from training, measured across a range of tasks, and are consistent with previous reports of their proficiency in math abilities (*Baron-Cohen et al., 2007*; *Iuculano et al., 2020*; *Iuculano et al., 2014*; *Jones et al., 2009*; *Oswald et al., 2016*; *Treffert, 2009*) and other cognitive skills (*Brown et al., 2010*; *Luyster and Lord, 2009b*; *Nemeth et al., 2010*; *Roser et al., 2015*). Children with ASD were able to improve on both accuracy and speed of problem solving. The more pronounced improvements on response time, relative to accuracy gain, as evident from larger effect sizes, may be reflective of children's response to our training that emphasized fluent problem solving as well as shift to memory-based strategy use. Further studies are needed to clarify how different types of training can enhance specific dimensions of problem-solving abilities in ASD.

Moreover, children with ASD also showed a positive transfer, reflected by faster reaction time on untrained problems, in contrast to TD children who did not show such generalized improvements following short-term training focused on trained problems. This finding may be surprising at first glance as it has been suggested that individuals with ASD may have a relative weakness in generalization due to an enhanced focus on details and surface features (*Happé and Frith, 2006*; *Harris et al., 2015*). However, training studies involving transitive interference and long-term memory have also failed to find generalization deficits in ASD (*Solomon et al., 2015*) and enhanced generalization in ASD has been reported for perceptual learning tasks (*Plaisted et al., 1998*). Our findings add to this literature and suggest that despite their potential weaknesses in social and other cognitive domains, learning and generalization in numerical problem solving in children with ASD, in some context, can be comparable or even superior to their TD peers. Although speculative, the lack of positive transfer to untrained problems in TD children may be a natural consequence of performance improvements on specific problems they practiced extensively, even at the cost of reduced performance on other problems that were not practiced. In this context, it is also possible that TD children may have formed expectations about performance on different types of problems (for example, better performance is expected for problems that are practiced more), whereas altered expectations about learning contexts may have blurred the distinction of expected peformance between different types of problems in ASD. Consistent with this view, a previous study has shown that individuals with ASD, relative to TD controls, are less surprised when their expectations are violated (*Lawson et al., 2017*). Follow-up studies are needed to further assess atypical expectations about performance in children with ASD. Taken together, our findings suggest that even a short-term, targetted training in numerical problem solving can induce near transfer or generalization in ASD, which could be one mechanism that contributes to enhanced math skills in some children with ASD (*Happé and Vital, 2009*).

## Cognitive mechanisms underlying learning in children with ASD

A second major finding of our study is that children with ASD acquired problem solving skills through distinct cognitive mechanisms from those of TD children. Specifically, while TD children relied on increased use of memory-based strategies following training, children with ASD were more likely to rely on rule-based strategies across both trained and untrained problems. These findings suggest that unlike TD children, who typically acquire efficiency in arithmetic problem solving by decreasing reliance on rule-based strategies and increasing use of memory-based strategies, children with ASD may develop efficiency in problem-solving procedures through continued reliance on rule-based learning. It is possible that children with ASD benefit from consistent adherence to explicitly instructed rule-based learning, in line with the perspective that affected individuals rely on higher-than-average systematizing mechanisms that focus on identifying regularities and rules (*Baron-Cohen and Lombardo, 2017*; *Baron-Cohen et al., 2003*).

However, a potential downside is that children with ASD may have relative weaknesses in flexibly switching between problem-solving strategies when it is advantageous to do so. Previous research has shown that stimulus repetition leads to inflexible learning in individuals with ASD, and reducing repetition can ameliorate adverse consequences in affected individuals (*Harris et al., 2015*). In the context of numerical problem solving, it is plausible that repeated practice with rule-based strategies may result in an overreliance of this strategy in children with ASD which persists beyond initial training, while TD children progress into memory-based strategies. Consistent with this view, it has been shown that individuals with ASD show increased influence of prior choices on subsequent decisions (*Feigin et al., 2021*). It is possible that while more efficient execution of rule-based strategies may contribute to preserved or enhanced generalization in children with ASD, persistence with same strategies may hinder learning in situations where flexible use of different problem-solving strategies is advantageous. An important direction for cognitive research in ASD is determining whether reduced repetition of the same strategy, and introduction of more flexible use of strategies, can enhance learning, similar to observations in perceptual learning (*Harris et al., 2015*).

It is important to note that heterogeneity in cognitive abilities and clinical symptomatology is a hallmark of ASD. Previous studies have consistently demonstrated that children with ASD exhibit a broad range of both cognitive strengths and challenges (*Chen et al., 2019*; *Fountain et al., 2012*; *Gotham et al., 2009*; *Gotham et al., 2012*; *Jones et al., 2009*; *Kjelgaard and Tager-Flusberg, 2001*; *Norbury and Nation, 2011*; *Wei et al., 2015*), which likely stem from their atypical learning styles. Heterogeneity in response to intervention may be also one of important features that differentiate children with ASD from their TD peers. Previous studies have pointed to the possibility of different types of atypical learning in ASD. One possibility is that individuals with ASD rely more on memorizing specific examples rather than grasping general patterns (*Qian and Lipkin, 2011*). Another possibility relates to hyper-systemizing traits in ASD, which are associated with enhanced systematic and logical thinking and learning abilities (*Baron-Cohen and Belmonte, 2005*; *Falter et al., 2008*). Findings from our study suggest that children with ASD may benefit from consistent adherence to explicitly instructed rule-based learning, which aligns with the perspective that affected individuals rely on higher-than-average systemizing mechanisms that focus on identifying regularizes and rules. While our study provides evidence for the atypical learning style associated with hyper-systemizing traits in children with ASD, it is possible that there are multiple subtypes of atypical learning within this population. Future studies with a larger sample of children with ASD may help determine the presence of subtypes of distinct learning styles in affected children. Our findings may also hold relevance for other neurodevelopmental disorders, including ADHD which are also known to be characterized by substantial heterogeneity (*Astle et al., 2022*; *Thapar et al., 2017*). Further research addressing comorbidities and transdiagnostic overlap is required to determine whether atypical learning patterns observed in the current study are specific to autism or they extend to other neurodevelopmental disorders.

In summary, although learning gains were comparable between children with ASD and TD children, detailed examination of problem-solving strategy use for trained and untrained problems allowed us to identify atypical cognitive mechanisms of learning in a domain of relative strength in many individuals with high-functioning ASD (*Happé and Vital, 2009*).

## Cognitive training, learning, and brain plasticity in ASD

The third major finding of our study is that neural mechanisms of learning are significantly different between children with ASD and TD children. Despite similar overall patterns of neural representational plasticity across the whole brain, including regions typically associated with math learning, our analysis of brain-behavioral relations revealed different patterns of relationship between NRP and learning between the two groups. Results from TD children revealed that training-related gains are accompanied by greater neural plasticity in brain systems associated with memory and quantity representation, instantiated in the MTL and IPS (*Arsalidou et al., 2018*; *Dehaene and Cohen, 1997*; *Menon, 2015*; *Wu et al., 2009*), as well as lateral occipital and frontal regions, a finding that is broadly consistent with previous reports (*Jolles et al., 2016*; *Rosenberg-Lee et al., 2018*; *Supekar et al., 2021a*). In contrast, in children with ASD, more stable neural representations in these same regions were related to better learning outcomes. Notably, atypical relations between brain plasticity and learning were observed in bilateral MTL, which further highlights the important role of MTL in math learning in TD children and altered hippocampal learning mechanisms in children with ASD. For

instance, changes in MTL activation and connectivity have been shown to be associated with response to training (*Chang et al., 2019*; *Rosenberg-Lee et al., 2018*) and longitudinal development (*Qin et al., 2014*) of memory-based numerical problem-solving skills in TD children. Our finding of atypical relation between MTL plasticity and learning in ASD is consistent with previous reports of abnormalities of the MTL in ASD (*Banker et al., 2021*; *Hogeveen et al., 2020*; *Schipul et al., 2012*; *Schumann et al., 2004*; *Welchew et al., 2005*). Moreover, such atypical relations were not observed in the MTL for problems that were not trained in children with ASD, confirming the functional specificity of our observations in training-induced learning.

Our finding that greater stability of neural representations is associated with greater learning in children with ASD has important implications in the context of persistence of rule-based strategies. We hypothesize that learning in children with ASD may be supported by stability of problem-solving strategies and neural representations across training sessions, which is broadly consistent with the notion that typical behavioral performance in ASD is achieved by atypical neural mechanisms (*Church et al., 2015*; *Dovgopoly and Mercado, 2013*; *Livingston and Happé, 2017*). As discussed above, these findings suggest that, unlike TD children, children with ASD may develop efficient problem-solving procedures through continued repetition and execution of rule-based strategies. While the flexible use of different problem-solving strategies or transitions between strategies is advantageous for TD children, children with ASD may show greater improvement using a less flexible approach.

Consistent with these hypotheses, we found a significant negative association between persistent use of rule-based strategy and NRP in the right middle frontal gyrus in children with ASD (*Figure 4—figure supplement 4*). Furthermore, moderation analyses (*Figure 5—figure supplement 1*) demonstrated that the association between lower NRP (i.e. increased stability in neural representation) and greater learning gains in the right FEF in children with ASD was driven by individuals who persistently used rule-based strategies across training sessions. In contrast, we did not observe such associations in TD children. These findings suggest that persistent use of rule-based strategy enhanced atypical patterns of brain-behavior relation in children with ASD. It is noteworthy that these findings were observed in prefrontal cortical regions implicated in visuospatial attention (*Ronconi et al., 2014*) and cognitive flexibility (*Sami et al., 2023*). Together, results from the current study highlight atypical patterns of problem-solving strategy use and brain plasticity that support preserved learning and identify neural sources of variability in cognitive skill acquisition in children with ASD in the context of educationally meaningful training.

It is worth noting that although, to the best of our knowledge, the present study is the first to probe the neurocognitive basis of individual differences in learning in children with ASD, our sample size was relatively modest. However, we used a rigorous experimental design with tightly controlled samples of children and included delivery of intended intervention and theoretically motivated analyses. Such approaches have been shown to enhance effect sizes for brain-behavior associations of interest (*Gratton et al., 2022*). Future investigations may further determine atypical brain-behavior relations in children with ASD across training contexts and cognitive domains. Together, our findings point to potential heterogeneous profiles of brain-behavior relations across brain systems implicated in learning and memory, quantity representation, visual perception, and cognitive control in children with ASD, and are consistent with previous observation that children with ASD do not engage the brain regions associated with math problem solving in a similar way as TD children (*Iuculano et al., 2020*).

## ASD symptomatology and learning-related brain plasticity

The final goal of this study was to investigate the role of RRIB symptomology on the relation between training-related learning and brain plasticity in children with ASD. We found that the relation between learning and plasticity of the MTL and IPS was moderated by insistence on sameness (IS), indicating that learning is supported by greater stability of neural representations in children with more severe IS symptoms. Furthermore, the moderating role of IS on the relation between learning and brain plasticity was not observed with respect to the two other RRIB components, circumscribed interests and repetitive motor behaviors. A possible explanation for this finding is that the IS construct has a clear conceptual overlap with cognitive and behavioral inflexibility and has been specifically linked to cognitive set shifting deficits (*Miller et al., 2015*). Our findings suggest that higher levels of IS in ASD may impair mental flexibility including the ability to make

strategy adjustments as a means of improving performance during learning. Moreover, previous studies have shown that IS is distinct from sensory-motor contributions to RRIB (*Miller et al., 2015*; *Mooney et al., 2009*). Our findings converge on these reports and provide evidence for the link between neural plasticity, learning, and the IS component of RRIB. Extending previous proposals of heterogeneity in learning in ASD (*Church et al., 2015*; *Mercado and Church, 2016*), our findings reveal IS as a key source of such heterogeneity and are consistent with a previous observation of less adaptable patterns of brain activity during learning that are associated with ASD symptoms (*Schipul and Just, 2016*). Together, these results suggest that a core phenotypic feature in ASD is linked to cognitive and behavioral inflexibility that contributes to atypical neural mechanisms of learning in affected children.

## Educational implications in children with ASD

Our findings have important implications for the development of effective pedagogical strategies in children with ASD. Given that students with ASD are increasingly being included in general educational and classroom settings (*Education, U. S. D. o, 2019*), heterogeneous cognitive and neural mechanisms of learning and their relation to clinical symptoms in affected individuals will need to be considered to establish more effective assessment and learning tools in classrooms and academic settings. Our study revealed a paradoxical persistence of rule-based strategies in children with ASD, even though shifting to a memory-based strategy generally represents a more efficient method for acquiring proficiency. On the one hand, such distinct characteristics underlying learning in children with ASD may be integrated into strengths-focused approaches to promote learning and self-confidence (*Cooper et al., 2021*; *Urbanowicz et al., 2019*). On the other hand, in contrast to TD children who naturally shift from rule-based to memory-based strategies once proficiency with rule-based strategies has been achieved, children with ASD may require more targeted training to transition to using memory-based strategies. It is noteworthy that explicit instructions for memory-based strategy provided in later training sessions in the current study were intended to mimic instructions in a typical classroom where children with ASD may attend. Although such an ecologically valid design can facilitate the investigation of the strategy shift in children with ASD in a classroom setting, future research is needed to examine how instructional methods and related social expectations may influence learning in affected children.

Also relevant to educational practice is our demonstration that IS moderates the relation between neural stability and learning gains in children with ASD. This novel clinically relevant finding suggests that optimizing learning in children with ASD will require special consideration of this core phenotypic feature of ASD. One pedagogical approach here might be to use pivotal response treatment to facilitate domain-specific cognitive skill development, along the lines of interventions used to improve social skills in children with ASD (*Koegel et al., 2019*). Precisely how individual interests, and cognitive strengths, or weaknesses, in individuals with ASD can be nurtured, or remediated, will be an important avenue for future research. More generally, we suggest that optimizing learning in children with ASD will require different strategies than in TD children, and educators and practitioners will need to take into consideration of how the unique and atypical features of learning in ASD, such as those identified in the present study, can be leveraged to maximize learning, academic performance, and overall quality of life (*Courchesne et al., 2020*; *Lord et al., 2020*).

There are several issues that require further consideration. First, larger sample sizes are required to further characterize heterogeneous patterns of atypical learning and whether the findings can be generalized to a broader ASD population. Second, our study focused on investigating atypical learning and associated neural mechanisms in children with ASD who had comparable cognitive abilities to our TD sample. Importantly, it remains unclear whether these mechanisms generalize to children with ASD with different levels of cognitive functioning, including children with lower IQ scores. It is hoped that future research addresses this question by including children with autism with a broader range of cognitive abilities. Third, further studies incorporating an appropriate active control intervention, in both children with ASD and TD groups, are needed to gain a better understanding of specific components of training that induce significant learning in children with ASD. Fourth, identifying brain mechanisms underlying cognitive inflexibility and its relation to cognitive skill acquisition in children with ASD is an important area for future research.

## Conclusions

In summary, our findings provide converging evidence that children with ASD have spared ability for acquiring mathematical problem-solving skills. Critically, learning in children with ASD is achieved by fundamentally different cognitive and neural mechanisms from their TD peers, providing novel support for the theory of atypical mechanisms of learning in children with ASD. We suggest that alterations in learning styles and brain plasticity may be one mechanism by which some children with ASD develop enhanced cognitive abilities. Our study points to distinct cognitive, neurobiological, and clinical features that contribute to variability in cognitive skill acquisition in ASD and provides a framework for establishing a more comprehensive understanding of individual differences in learning in childhood autism.

# Materials and methods

## Participants

A total of 116 children were recruited from the San Francisco Bay Area via flyer or poster advertisements at schools, libraries, and community centers. The informed written consent was obtained from the legal guardian of each child and all study protocols were approved by the Stanford University Review Board (IRB-11849). All participants were volunteers and were treated in accordance with the American Psychological Association 'Ethical Principles of Psychologists and Code of Conduct'. All children were right-handed, 8–11 years of age (grades 3–5), had a full-scale IQ>80 based on the Wechsler Abbreviated Scale of Intelligence (WASI) (*Wechsler, 1999*), and had no history of claustrophobia and previous head injury. The diagnosis of ASD was based on DMS-IV and the Autism Diagnostic Interview-Revised (ADI-R) (*Lord et al., 1994*) and/or the Autism Diagnostic Observation Schedule (ADOS) (*Luyster et al., 2009a*), and confirmed by an experienced clinical psychologist. Additional inclusion criteria for TD children were had no history of genetic, neurological, psychiatric, or learning disorders, no personal and family history (first degree) of developmental disorders, and no significant difficulty during pregnancy, labor, delivery, or immediate neonatal period, or abnormal developmental milestones.

All children completed neuropsychological assessments and their parents or legal guardians completed demographics, medical history, screening for MRI scan safety, and questionnaires regarding their child. In addition, children with ASD completed diagnostic assessments to confirm their autism diagnosis. Children's eligibility to participate in the training study was determined by neuropsychological assessment scores (FSIQ>80), ability to perform MRI protocol procedures, and autism diagnostic criteria. Twenty-one children did not meet the inclusion criteria and were not eligible to participate in the training study. Among eligible participants, 23 children discontinued due to a lack of interest or difficulties in coordinating their schedules. Eight children were unable to continue with the study due to challenges in following instructions during the sessions. Finally, one child was excluded due to task administration error. A total of 35 children with ASD and 28 TD children who completed training and pre/post behavioral and brain imaging sessions were included in the current study. A detailed visual depiction of participant inclusion procedures is provided in *Figure 1—figure supplement 1*. All eligible participants completed the training protocol in this study. One child with ASD was excluded due to the lack of valid data in the fMRI task, with an accuracy below chance level (<50%) and no responses to over 30% of trials in more than half of runs in the pre-training fMRI task. The final sample consisted of 35 children with ASD (age=9.98±0.92; 29 boys; IQ=117.71±15.72) and 28 age-, gender-, IQ-matched TD children (age=10.00±1.09; 22 boys; IQ=118.64±9.41) (*Supplementary file 1*). This sample size was consistent with the power analysis from our pilot study, which suggested that a group size of 25 is sufficient for identifying behavioral changes with training in children with ASD, with a power>80% and an effect size of Cohen's $\delta$=0.9. The number of children for subsequent analysis varied based on available high-quality behavioral and/or fMRI data for each analysis (*Supplementary file 2*).

## Study design and procedure
### Overall study protocol
The overall study protocol is summarized in *Figure 1a*. This study consisted of the following sessions: (*i*) pre-training neuropsychological (NP) assessments of cognitive abilities and clinical assessments; (*ii*)

pre-training task fMRI scan session (math verification task) and outside-of-scanner assessment session (math production task and strategy assessment); (*iii*) one-on-one math training, in which 5 training days were spread out across a 2-week period, with no more than 3 days between training days; (*iv*) post-training task fMRI scan session and outside-of-scanner assessment sessions with the same tasks as pre-training.

## Training problem sets

Both children with ASD and TD children were randomly assigned to one of two training problem sets, Set A and Set B. Each training set consisted of 14 double-digit plus single-digit problems (*Figure 1— figure supplement 2*). The same training problem set was used across tasks in behavioral and neuroimaging sessions before, during, and after 5 days of training. The problems (28 in total) were counterbalanced between Set A and Set B, with problems used as training (trained) problems in one set were used as novel (untrained) problems for the other set. The problems were generated by the following a well-developed procedures, which have been used to create well-matched problem sets in our previous study with TD children (*Chang et al., 2019*). Specifcally, all possible single-digit addition problems with operands from 2 to 9 were created, excluding ties (e.g. 2+2). Half of 28 problems had the smaller operand first, and the other half had the larger operand first. These problems were further divided into two-well balanced sets of 14 problems with the sum of each set equal to 154. Each set had 6 'non-carry over' problems with sums of 10 or less and 8 'carry over' problems with sums of 11 or more. Half of the problems in each set were randomly assigned to have double-digit in the first operand (or double-digit in the second operand), with decade values in the double-digit operands ranging from 20 to 80. The assignment of decade values to single-digit problems amongst each set of 14 problems had to follow several constraints: (*i*) the numbers 2–9 appear at least once as the single-digit operand; (*ii*) double-digit multiples of 11 are excluded; (*iii*) at least one problem is summed to a value in each of the decades from 20 to 90; (*iv*) sums are separated by at least three units. Finally, all sums were unique across the training sets.

## Training activities

All children participated in five days of training with a tutor. Our math training protocol was designed to gradually transition from intensive practice with problem-solving procedures to the use of the memory-retrieval strategy by the end of training. Children completed 8 interactive activities that involved solving each of 14 trained problems 15 times, thereby repeating 75 times in total across 5 training days. On each training day, the tutor first introduced a 'break-apart' (decomposition) strategy to children to facilitate their learning of complex arithmetic problem solving, after which children completed a worksheet to solve a set of trained problems using the break-apart strategy. Children were encouraged to use the memory-retrieval strategy whenever possible on the last day of training. Below is the list of activities that children were engaged in each day of training (Days 1–5). Participants accumulated stickers for completing each activity on a 'treasure board' and were invited to select a small prize upon completion of 20 training activities. Sample training materials are shown in *Figure 1a*.

i. Warm-up flashcard. The first task each day, except for Day 1 of training, was a simple warm-up flashcard task, in which participants were shown each of 14 problems on a flashcard and were asked to verbally produce the solution to the problem. The tutor proceeded to the next problem once the child produced a correct answer.

ii. Lesson and worksheet. After warm-up, participants were given an introductory lesson on 'break-apart' (decomposition; rule-based) strategy, which involved separating the double-digit operand into a decade (multiple of 10) and a single-digit number and then adding the sum of the two single-digit numbers to the decade number (e.g. 65+7 => 60+5 + 7=60 + 12=72). On a 14-problem worksheet, the tutor demonstrated this method on the first problem, then asked the child to solve the next one with guided support. Once participants understood the strategy, they were prompted to complete the worksheet using the break-apart strategy. While participants were asked to use the break-apart strategy for the first four days of training, on Day 5, participants were encouraged to solve the problems by retrieving the answers from memory (memory-retrieval strategy).

iii. Treasure hunt/Bingo. Participants played a multiple-choice game in which they were asked to match a presented solution to a problem to one of the problems on the board. There were two versions of this game, 'Treasure hunt' (played on Days 1, 3, and 5) and 'Bingo' (played on

Days 2 and 4). In 'Treasure hunt', participants were shown a treasure map containing 14 boxes with problems and a deck of 14 cards with possible solutions placed faced down. Participants drew one card at a time and placed it on one of the boxes in the treasure map to match the solution (card) to the problem (box). In 'Bingo', participants received a 4x4 bingo card, where each square (except for two 'free spots') contained a problem. Participants were given bingo chips with solutions to match to corresponding problems on the card.

 iv. Pirate game (computer). Participants played three rounds of a computerized flashcard game in which each of 14 problems was presented in a random order on the screen in an animated soap bubble. Participants completed each problem by typing the answer to the problem and pressing the enter key. Each bubble contained a trained problem appeared in the bottom of the screen and moved upwards until it reached the top of the screen. When the participant answered the problem correctly within the time the bubble remained on the screen, the bubble popped and the next problem appeared at the bottom of the screen. Multiple attempts were allowed to enter the correct response before the time ran out when the bubble reached the top of the screen and the next problem appeared at the bottom of the screen. The time limit was set as 15 s for the first round of the Pirate Game in the first training session and was reduced by one second after each successful round (>75% accuracy) across training days, reaching 9–11 s for the final round of the last training day.

 v. Oral review. Participants were asked to verbally produce the answer to each of 14 problems read out loud by the tutor, without the use of worksheet or physical manipulatives. The order of problems presented varied from day to day. All participants in the same group (children assigned to training set A or B) received the same order.

 vi. Memory game. Participants completed a memory game using fill boards from the game 'Guess Who' (Hasbro Gaming, Pawtucket, Rhode Island) boards. The 14 problems were split into two boards with seven pairs of problem and answer on each. Problems were distributed on the top two rows and solutions on the bottom two rows. All cards were initially faced down and participants were instructed to pick one problem and one solution to see if they matched. If they matched, these cards would remain face up and if they did not match, they would be turned face down and new pairs would be picked until all problems and solutions were matched. The placement of the pairs remained constant throughout the days of training and all participants completed the two memory boards in the same order. The tutor timed the duration taken to complete each board for each participant. In addition, they were encouraged to find matching problem-solution pairs in as few moves as possible to elicit memory retrieval for spatial locations.

 vii. Beat your time. Participants completed three rounds of timed flashcard game in which a deck of flashcards containing 14 problems was completed twice in each round. The tutor presented each card to the participant one at a time. Participants verbally produced correct answer to move onto the next problem. If an incorrect answer was produced, the tutor waited until the participant provided the correct answer before continuing. The tutor timed the duration taken to complete each round and participants were encouraged to beat the time taken to complete previous round.

 viii. Review worksheet. Participants completed a 14-problem worksheet at the end of every training session for review. Each day, participants were prompted to solve the problems however easiest for them. By Day 5, children were encouraged to use the memory-retrieval strategy.

To characterize their learning profiles, children's performance on each training day was recorded in the computerized training task, 'Pirate Game'. We computed a composite inverse efficiency score (IES) by dividing mean reaction time by accuracy (*Bruyer and Brysbaert, 2011*). The mean reaction time for each child on each training day was calculated using correctly solved trials. Children's learning rate was obtained by fitting daily measures of inverse efficiency score in a linear regression model.

## Behavioral tasks

### Math verification task in the fMRI scanner

Before and after training, children completed a math verification task in the scanner. As shown in *Figure 1a*, each trial began with a fixation cross with 500ms, followed by a double-digit plus single-digit problem presented for 6 s. During this problem presentation phase, participants were instructed to solve the problem. Next, a possible solution to the problem (probe) was presented for up to 3 s. During this response phase, participants indicated whether the possible solution was same or different from the answer to the problem they were thinking of by pressing the left button with their index

finger or the right button with their middle finger, after which a blank screen filled a 10-s trial length followed by a jitter period ranging from 8 to 12 s.

Participants completed a total of 4 runs. Each run consisted of 7 trained and 7 untrained problems, drawn from 14 trained and 14 untrained problems. Each of all 28 problems was presented twice across the 4 runs, once in the first two runs, and for the second time in the last two runs. In each run, half of the probes presented were correct answers and the other half were incorrect. Within each run, correct and incorrect problems were presented in a pseudo-random order where no more than three correct or incorrect problems appeared in a row. Incorrect answers differed from the correct answer by plus or minus 1, 2, or 10. All possible differences from the incorrect sum were used once in each run, as well as one trial presented an incorrect answer of either +1 or –1 for a total of 7 incorrect answers per run. If the first run had an extra +1 trial, then the second run had the –1 trial and vice versa. Notably, answers differing by plus or minus 10 ensured that children considered both operands to solve the problems. Run were not repeated even in cases of excessive head movement, in order to ensure that all participants have the same amount of exposure to the problems.

Given that reaction time was collected only during the response phase in this task design, performance was assessed by accuracy (ACC) averaged across runs for trained or untrained problems at pre- and post-training. Runs with an accuracy below chance level (<50%) and no responses to 30% or more trials were considered invalid and were not included in the analysis. To assess training-induced learning controlling for baseline performance on this task, learning gains in this task were measured as percent changes in accuracy from pre- and post-training ($ACC_{gain}$ = ($ACC_{post}$ - $ACC_{pre}$)/$ACC_{pre}$).

## Math production task and strategy assessments

After the math verification task in the fMRI scanner, children completed an out-of-scanner task that included arithmetic production ('math production task') and strategy assessments, consisting of the same 14 trained and 14 untrained problems. The problems were presented in the same order in each of pre- and post-training sessions across all children, with the order of the problems reversed between pre- and post-training sessions. In the math production task, children were required to solve each addition problem by verbalizing their answers. The time taken to solve each problem was recorded via a button press by a trained assessor. There was no time limit for the task, and on average, most problems were solved correctly even before training (see *Supplementary file 4* for details); therefore, children's performance on this task was assessed by reaction time (RT) for correctly solved trained or untrained problems. Although there was no time limit for this task, changes in RT for trained problems were expected following our short-term training protocol that emphasized the use of memory-based strategy associated with faster problem solving. Similarly, learning gains in this task were measured as percent changes in reaction time from pre- and post-training ($RT_{gain}$ = ($RT_{post}$ - $RT_{pre}$)/$RT_{pre}$).

After solving each problem, children's problem-solving strategies were assessed ('strategy assessments'), in which they were asked to describe how they solved the problem (*Wu et al., 2008*). Children's responses were recorded verbatim. Based on the observation of overt strategies (e.g. finger counting) and children's verbal responses, the assessor coded one or more problem-solving strategies, including rule-based strategies, such as counting and decomposition (e.g. "I broke down 8–3 and 5 and did 75 plus 5, and then added 3"), and memory-based retrieval strategy (e.g. "I remembered the answer to 75 plus 8"). Children's dominant strategy was determined by the most frequently used strategy across correctly solved problems in each trained and untrained condition at pre- and post-training.

### Behavioral data analysis

### Computerized task (Pirate Game) during training

For computerized task administered on each day of training, we performed a 5x2 repeated measures ANOVA on inverse efficiency score with sessions (Days 1, 2, 3, 4, 5) as a within-subject factor and group (ASD, TD) as a between-subject factor. A one-way repeated measures ANOVA was performed on inverse efficiency score across sessions (Days 1, 2, 3, 4, 5) in each group. For the learning rate derived from a linear regression model (i.e. $y=ax + b$), a two-sample *t*-test was used to examine the differences between groups.

## Math verification and production tasks

For both of verification and production tasks, a 2x2 repeated measures ANOVA with time (pre-training, post-training) as a within-subject factor and group (ASD, TD) as a between-subject factor was performed. To further assess group differences in performance for each task, we conducted planned two-sample $t$-tests.

## Strategy assessments

A Chi-squared test was used to examine dominant strategy (rule-based or memory-based) used for trained problems and strategy differentiation between trained and untrained problems at pre- and post-training. Specifically, we examined the differences in the distribution of dominant strategy used between groups (ASD, TD) for trained problems. Then, we further examined the differences in the distribution of dominant strategy used between problem types (trained, untrained) in each group. Cohen's $d$ (for $t$-tests), $\eta^2_p$ (for ANOVA), or $\phi$ (for Chi-squared tests) (*Cohen, 2013*) was calculated to provide estimates of effect sizes. Additionally Bayes factors (BF) were computed to assess the relationship between behavioral changes on each measure in two groups (*Rouder et al., 2012*; *Rouder et al., 2009*; *Wagenmakers et al., 2010*). In the context of Bayesian inference, the null hypothesis (H0) assumes no group differences in behavioral changes, while the alternative hypothesis (H1) would support the presence of group differences. Values greater than 10 in favor of H1 provide strong evidence supporting H1, values between 3 and 10 provide moderate evidence, values between 0.33 and 3 provide insufficient evidence, values between 0.10 and 0.33 in favor of H0 provide moderate evidence of absence, and values below 0.10 provide strong evidence of absence.

## fMRI data acquisition

fMRI data were acquired on a 3T GE Signa scanner (General Electric, Milwaukee, WI) using a custom-built 8-channel head coil at the Richard M. Lucas Center for Imaging at Stanford University. Head movement was minimized during the scan by placing cushions around the participant's head. During the fMRI task scanning, children held a custom-made MR-compatible computer mouse in their right hand. A total of 31 axial slices (4.0 mm thickness, 0.5 mm skip) parallel to the anterior commissure (AC)-posterior commissure (PC) line and covering the whole brain were imaged using a T2*-weighted gradient-echo spiral in-out pulse sequence (*Glover and Lai, 1998*) with the following parameters: repetition time (TR)=2 seconds, echo time (TE)=30ms, flip angle=80°, one interleave. The field of view was 22 cm, and the matrix size was 64×64, providing an in-plane spatial resolution of 3.4375 mm. Reduction of blurring and signal loss arising from field inhomogeneity was accomplished by the use of an automated high-order shimming method based on spiral acquisitions before fMRI acquisition (*Kim et al., 2002*). Each task fMRI run lasted 4 min and 50 s (i.e. 145 volumes/time points) including 10 s at the beginning of each run for allowing scanner equilibration.

## fMRI data preprocessing

fMRI data were analyzed with the following preprocessing procedures using SPM12 (https://www.fil.ion.ucl.ac.uk/spm/). The first 5 volumes (10 s) were not analyzed to allow for signal equilibration. A linear shim correction was applied separately for each slice during reconstruction based on a magnetic field map acquired automatically by the pulse sequence at the beginning of scan (*Glover and Lai, 1998*). The subsequent processing included the motion correction with realigning to the first scan and the slice-timing correction. Then, images were spatially normalized to standard Montreal Neurological Institute (MNI) space using the echo-planar imaging template, resampled to 2 mm isotropic voxels using trilinear sinc interpolation, and smoothed with a 6 mm full-width at half-maximum Gaussian kernel. Data from 13 children with ASD and 4 TD children were excluded in the following analysis due to excessive head motion (translation or rotation above 10 mm or 10° in any direction or mean frame-wise head motion above 0.5 mm) in more than half of runs. Data from runs with excessive head motion were also excluded for the following analysis. Additionally, as part of quality control of acquired data, the image intensity at the pre-training session from one child with ASD was identified as an outlier (>3 standard deviations away from the mean) and excluded from data analysis. A gray matter mask with 172,470 voxels was generated by overlapping image mask across children and SPM gray matter template (grey matter probability >0.2) to limit our analysis to the gray matter. Given that

some participants did not have full coverage of the cerebellum and brainstem, our analysis focused on cerebral regions.

## First-level statistical analysis

Task-related brain activation was identified using a general linear model based on the preprocessed image data during the problem presentation phase (6 s) in the math verification task. For each child, brain activity representing correct and incorrect trials for each trained and untrained condition – a total of four conditions – was modeled using boxcar functions with a canonical hemodynamic response function and a temporal dispersion derivative to account for voxel-wise latency differences in hemodynamic response. Six movement parameters estimated by the realignment procedure were included as regressors of no interest. A high-pass filter (0.5 cycles/min) was used to remove the low-frequency drifts for each voxel. Serial correlations were accounted for by modeling the fMRI time series as a first-degree autoregressive process. Voxel-wise $t$-statistics maps contrasting correct trials for trained and untrained problems versus baseline were generated for each child.

## Multivariate neural representational pattern analysis

To characterize training-related functional brain plasticity on a fine spatial scale that extends beyond canonical univariate analysis methods, we performed multivariate neural representational pattern analysis (*Kragel et al., 2018*; *Kriegeskorte et al., 2008*; *Supekar et al., 2015*) and obtained *neural representational plasticity* (NRP) measure (*Figure 1b*). First, within a 6 mm spherical region centered at each voxel, we calculated spatial Pearson's correlation of brain activation ($t$-scores) between pre- and post-training for trained condition. Our analysis focused on problems that were correctly solved. NRP was then obtained by normalizing Pearson's correlation coefficient using Fisher's $r$-to-$Z$ transformation ($Z=0.5*\ln((1+r)/(1 r))$) and multiplying this value by –1. Higher NRP scores indicated the degree of training-induced functional brain plasticity. The NRP scores were assigned to the voxel and this procedure was repeated for all voxels across the whole brain, using searchlight mapping (*Kriegeskorte et al., 2006*).

### Group differences

To examine whether children with ASD and TD children show similar or different patterns of functional brain plasticity in response to training, individual whole-brain NRP maps were submitted to two-sample $t$-test. Anatomical brain locations of significant clusters were identified using Harvard-Oxford atlas (*Desikan et al., 2006*) and Juelich Histological atlas (*Eickhoff et al., 2005*).

### Relation between NRP and learning gains

Next, to examine whether the relationship between functional brain plasticity and learning are similar or different between ASD and TD groups, a general linear model was used with group (ASD, TD), learning gains, and their interaction as independent variables, and NRP as the dependent variable. Here, learning gains were measured by percent changes in accuracy for trained problems in the math verification task, considering that this task was administered during fMRI scanning and thus provides performance/learning related measure directly associated with brain activation/plasticity. Additionally, to examine whether the observed findings are specific to trained problems, we performed similar analyses for the untrained problems.

Significant clusters were identified using a voxel-wise height threshold of $p<0.005$ and an extent threshold of $P<0.05$ using family-wise error correction for multiple comparisons based on GRF. Cohen's $f$ was calculated to provide estimates of effect sizes. In addition to the whole brain analysis, we further examined NRP for trained problems within a priori MTL and IPS regions strongly implicated in learning and memory and quantity representation (*Butterworth and Walsh, 2011*; *Menon, 2016*; *Menon and Chang, 2021*; *Figure 3b*).

## Influence of clinical symptoms on the relationship between brain plasticity and learning

To examine how restricted and RRIB influence the relationship between brain plasticity and behavioral changes following math training in children with ASD, moderation analyses were conducted in R [Version 4.0.2, 2020] (*R Development Core Team, 2013*). Three distinct subcomponents of RRIB were

calculated based on scores from ADI following the procedures described in a recent study (*Supekar et al., 2021b*): insistence on sameness, circumscribed interests, and repetitive motor behavior. Circumscribed interests and insistence on sameness subscores capture children's adherence to specific interests and routines and are considered to be cognitive components of RRIB. In contrast, the repetitive motor component encompasses actions such as hand flapping, rocking, and head banging, behaviors which are considered motoric in origin (*Lewis et al., 2007*; *Turner, 1999*). For each RRIB component, learning gains were entered as the dependent variable, NRP was entered as the independent variable, and RRIB component scores were entered as the moderator variable. NRP was estimated in the left and right MTL and right IPS regions identified from interaction between group and learning in whole brain analysis.

## Code availability

Data were analyzed using MATLAB R2018b, SPM12, DPABI, and R. Data analysis scripts are available at https://github.com/scsnl/jinliu_ASD_plasticity_2022 copy archived at *Liu, 2023*.

## Acknowledgements

This research was supported by the United States National Institutes of Health to VM (HD059205, MH084164, HD094623) and MR-L (MH101394), and by the Stanford Maternal & Child Health Research Institute Postdoctoral Support Award to HC and JL. We thank participating families, Jennifer Phillips, Kaustubh Supekar, Yuan Zhang, and Holly Wakeman for assistance with the study.

## Additional information

### Funding

| Funder | Grant reference number | Author |
| --- | --- | --- |
| National Institutes of Health | HD059205 | Vinod Menon |
| National Institutes of Health | MH084164 | Vinod Menon |
| National Institutes of Health | HD094623 | Vinod Menon |
| Stanford Maternal and Child Health Research Institute | | Jin Liu Hyesang Chang |
| National Institutes of Health | MH101394 | Miriam Rosenberg-Lee |

The funders had no role in study design, data collection and interpretation, or the decision to submit the work for publication.

### Author contributions

Jin Liu, Conceptualization, Data curation, Formal analysis, Funding acquisition, Investigation, Visualization, Methodology, Writing – original draft, Writing – review and editing; Hyesang Chang, Conceptualization, Data curation, Formal analysis, Funding acquisition, Validation, Investigation, Writing – original draft, Project administration, Writing – review and editing; Daniel A Abrams, Conceptualization, Supervision, Investigation, Writing – review and editing; Julia Boram Kang, Data curation, Project administration, Writing – review and editing; Lang Chen, Data curation, Methodology, Project administration, Writing – review and editing; Miriam Rosenberg-Lee, Conceptualization, Data curation, Project administration, Writing – review and editing, Funding acquisition, Supervision; Vinod Menon, Conceptualization, Resources, Supervision, Funding acquisition, Investigation, Methodology, Writing – review and editing

### Author ORCIDs

Jin Liu ⬚ https://orcid.org/0000-0003-4343-2623

Hyesang Chang http://orcid.org/0000-0002-2231-1112
Daniel A Abrams https://orcid.org/0000-0002-1255-1200
Lang Chen http://orcid.org/0000-0002-2118-5601
Miriam Rosenberg-Lee http://orcid.org/0000-0003-0768-0347
Vinod Menon http://orcid.org/0000-0003-1622-9857

## Ethics

Human subjects: The informed written consent was obtained from the legal guardian of each child and all study protocols were approved by the Stanford University Review Board (IRB-11849). All participants were volunteers and were treated in accordance with the American Psychological Association 'Ethical Principles of Psychologists and Code of Conduct'.

## Decision letter and Author response

Decision letter https://doi.org/10.7554/eLife.86035.sa1
Author response https://doi.org/10.7554/eLife.86035.sa2

# Additional files

## Supplementary files

• Supplementary file 1. Demographic and clinical measures.

• Supplementary file 2. Number of participants included in each analysis.

• Supplementary file 3. Results of repeated measures ANOVA for behavioral performance.

• Supplementary file 4. Results of t-tests for behavioral performance.

• Supplementary file 5. Results of chi-squared tests for dominant strategy use.

• Supplementary file 6. Brain regions showing significant group by behavior interaction on neural representational plasticity between pre- and post-training for trained problems.

• Supplementary file 7. Results of brain-behavior association between region of interest (ROI)-based neural representational plasticity and learning gains.

• Supplementary file 8. Moderation results for RRIB sub-scores on the association between brain and behavioral measures.

• MDAR checklist

## Data availability

The dataset for this article has been uploaded to the National Institute of Mental Health Data Archive (NDA) and is freely available to all qualified researchers upon submission of an access request (https://nda.nih.gov/edit_collection.html?id=2628). Data usage should adhere to the NDA Data Use Certification. Raw datasets for this article are not made publicly available due to agreement in the Institutional Review Board approved informed consent with study participants. Reasonable requests to access raw datasets should be directed to Dr. Vinod Menon (menon@stanford.edu), following the study protocol approved by the Stanford University Institutional Review Board. SPM12, used for task-related fMRI data preprocessing and analysis, is openly available at http://www.fil.ion.ucl.ac.uk/spm/. The code and de-identified processed dataset used for this study are publicly available on GitHub at https://github.com/scsnl/JL_eLife_2023 (copy archived at *Stanford Cognitive & Systems Neuroscience Laboratory, 2023*) and https://osf.io/5ywrt/.

The following datasets were generated:

| Author(s) | Year | Dataset title | Dataset URL | Database and Identifier |
|---|---|---|---|---|
| de los Angeles C | 2023 | JL_eLife_2023 | https://doi.org/10.17605/OSF.IO/5YWRT | Open Science Framework, 10.17605/OSF.IO/5YWRT |
| Menon V | 2017 | Learning and Brain Plasticity in Children with Autism: Relation to Cognitive Inflexibility and Restricted-Repetitive Behaviors | https://nda.nih.gov/edit_collection.html?id=2628 | NIMH Data Archive, 2628 |

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
