## [Editor Report]

This is an important study on learning strategy differences in autism vs typically developing controls. The study identifies similar learning rates but different learning strategies. The evidence provided by the authors is compelling, relying on well-done tasks and fMRI analyses. This paper will be of broad interest to the autism community and to the study of learning.

---

## [Decision Letter]

**Decision letter after peer review:**

Thank you for submitting your article "Atypical cognitive training-induced learning and brain plasticity and their relation to insistence on sameness in children with autism" for consideration by *eLife*. Your article has been reviewed by 3 peer reviewers, one of whom is a member of our Board of Reviewing Editors, and the evaluation has been overseen by Floris de Lange as the Senior Editor.

Essential revisions:

1) Please discuss generalization to autism more broadly defined given the small sample size and the sample characteristics (high IQ, etc.)

2) Further analyses of strategy independent of diagnosis on brain imaging measures (see reviewer #2).

3) Answer the comments on efficiency measures by reviewer #2

*Reviewer #1 (Recommendations for the authors):*

Statistics: the results show, in addition to t or F and p values, BF; I assume these are Bayes Factors? They are never explained in the methods. And if they are indeed Bayes Factors then some very low values (i.e. 0.32) are associated with insignificant results, which would be a weird interpretation.

Behaviour statistics: explain how multiple comparisons were accounted for.

Intro paragraph: ASD and intellectual disability are contrasted as two separate things, yet often co-occur. Maybe moderate the language a little to reflect that.

The acronym BF needs to be explained and contextualized.

Intro to methods: be clearer about how the initial n=116 was reduced to 35 ASD and 28 TD.

Throughout the text, especially in the discussion: make it clearer whether they expect a unitary autism learning paradigm, or subsets of children with different learning strategies. That is, explore the role of the wide heterogeneity of the condition on learning. Also some discussion of whether they would expect this to be particular to autism or generalize to related neurodevelopmental disorders. (I realize that there is some discussion about IS as the mediating factor to explain heterogeneity, but I believe that there could be further emphasis and discussion throughout the text)

*Reviewer #2 (Recommendations for the authors):*

Overall I commend the authors for a nicely conducted intervention study, a clear and elegant series of analyses, and a compelling description of what is largely a null result (i.e., nearly identical training and NRP changes across groups). I think this is particularly timely, as there are a growing number of findings from our group and others that suggest many critical cognitive processes may be implemented differently, but remain relatively 'spared' in ASD relative to TD.

To my reading, there are four primary weaknesses of this paper in its current form:

1. The efficiency score does not seem to be calculated according to what is most often done in this literature. Typically, reaction time is divided by 1 minus the proportion of errors, and higher scores indicate LESS efficient performance (in the current paper, the key behavioral training efficiency was operationalized using accuracy divided by reaction time, with higher scores indicating MORE efficient performance). While these measures will be monotonically-related, based on some of my own back-of-the-envelope math as well as a series of formalizations in Bruyer and Brysbaert (2011; https://citeseerx.ist.psu.edu/document?repid=rep1andtype=pdfanddoi=1fcafd7a66caad684ae311d7468f4ca451c8dd0a), it seems likely that they may lead to distinct inferences when the proportion of errors are relatively large (e.g. >20%, as they are for several subjects here), or if there is not a high association between errors and response latency across subjects. I would appreciate it if the authors would either recalculate their efficiency measures so it is more in line with the formula that is commonly used in the literature [IES = RT / (1-prop error)]. Or, if the authors know there is precedent for the present formula that I am unaware of (quite possible!), some consideration of whether the current data meets the assumptions that must be met to support the use of the efficiency score in the first place (i.e., error rates largely <10-15%, between-subjects correlations between speed and accuracy, etc.) would be appreciated.

2. The current results comprise an ASD sample that has a mean IQ that is above the mean population IQ. This has limited generalizability to a significant proportion of individuals with ASD. This is common in the field, and I am sensitive to the logic behind IQ-matched samples, but some discussion of it as a limitation in the Discussion seems appropriate.

3. The study does not contain an adequate active control or sham condition to adequately account for non-intervention-related variance in behavior and BOLD data across pre-to-post test. Therefore, it is impossible to know what the active ingredients of the intervention are. This was not a core aim of the work, so not a huge deal, but it would be good to, again, consider it as a potential limitation to be followed up in future studies.

4. The authors argue that their behavioral and neural results may reflect the use of different dominant strategies to perform problem-solving tasks in ASD relative to TD. However, at this stage the dominant strategy models appear to be largely independent of the behavioral performance, NRP, and clinical inflexibility variables, making it hard to fully interpret them relative to the other data in the paper. The authors might want to conduct additional analyses to support this claim, for example, testing whether the strategic differences played any functional role in shaping the behavioral and neural changes from the pre-post test within or between groups. For example: does the degree to which individuals with ASD stuck to a rule-based strategy modulate the associations observed in Figures4 or 5? Is this similar or different to the impact of strategy on brain-behavior relationships in TD? etc.

*Reviewer #3 (Recommendations for the authors):*

The discussion of savants and exceptional abilities in the introduction was interesting and it may be beneficial to include such information in the analyses or as part of the participant information section. For instance, were there any autistic individuals in the study with exceptional skills and did they then also rely on the same memory-based strategies as the average? It may therefore be beneficial to include an individual approach or description.

The investigation of insistence on sameness as a moderator of brain plasticity and learning was a valuable analysis. Did the authors consider a brain region as a potential moderator or mediator such as the insula or other behaviors such as cognitive flexibility/shifting? Additionally, was a total RRIB score used in any of the analyses instead of individual RRIBs? Or perhaps elaborate on why different forms of RRIBs were not significant. Further, why were RRIBs used rather than the score from the ADOS RRB?

The finding that learning is supported by greater stability of neural representations in autism could be further discussed.

"Data from 13 children with ASD and 4 TD children were excluded in the following analysis due to excessive head motion (translation or rotation above 10mm or 10{degree sign} in any direction or mean frame-wise head motion above 0.5mm) in more than half of runs. Data from runs with excessive head motion were also excluded for the following analysis." More information about the head motion profiles for subjects would be useful, including if there were significant differences in head motion between the groups. And this information may be included in supplementary File 1.

A further description of the RRIB measures, such as what each of them captures, could be included in the method or supplementary materials beyond the 2021 reference in the methods section.

Figure 1: More description of the tasks could be included in the figure legend. Further, details on what fMRI scan was used (pre v post-training) should also be included.

Figure 2: It appears that there is a near-equal split between rule-based and memory-based strategies used in the ASD group. This could be further clarified in the results or perhaps emphasized – regarding greater reliance on rule-based methods and the untrained group (Figure 2d).

Figure 4: Unfortunately the text is too small and blurry to really assess this figure. However, it appears that the blue line (ASD group) is declining. However, in the manuscript, it is stated that the neural changes are "stable". Please clarify/emphasize the use of the word "stable" in relation to the plots.

---

## [Author Response]

Essential revisions:1) Please discuss generalization to autism more broadly defined given the small sample size and the sample characteristics (high IQ, etc.)2) Further analyses of strategy independent of diagnosis on brain imaging measures (see reviewer #2).3) Answer the comments on efficiency measures by reviewer #2

Thank you for highlighting these essential comments from the reviewers. We have revised the manuscript according to the reviewers’ suggestions, as described in detail below.

Reviewer #1 (Recommendations for the authors):Statistics: the results show, in addition to t or F and p values, BF; I assume these are Bayes Factors? They are never explained in the methods. And if they are indeed Bayes Factors then some very low values (i.e. 0.32) are associated with insignificant results, which would be a weird interpretation.

We thank the reviewer for raising this important point. In addition to statistical inference based on frequentist *p* values, we also conducted Bayesian analysis for inference of the absence or presence of evidence. The strength of the evidence is characterized by examined Bayes Factor (BF)(Rouder et al., 2012; Rouder et al., 2009; Wagenmakers et al., 2010). Importantly, BF values between 0.10 and 0.33 in favor of H0 provide moderate evidence of absence, and therefore the example cited by the reviewer (i.e., 0.32) is consistent with an insignificant finding. We have now provided detailed explanation about the BF and associated statistical inference in the Methods section (Page 21) in the revised manuscript as follows.

“Additionally Bayes factors (BF) were computed to assess the relationship between behavioral changes on each measure in two groups (Rouder et al., 2012; Rouder et al., 2009; Wagenmakers et al., 2010). In the context of Bayesian inference, the null hypothesis (H0) assumes no group differences in behavioral changes, while the alternative hypothesis (H1) would support the presence of group differences. Values greater than 10 in favor of H1 provide strong evidence supporting H1, values between 3 and 10 provide moderate evidence, values between 0.33 and 3 provide insufficient evidence, values between 0.10 and 0.33 in favor of H0 provide moderate evidence of absence, and values below 0.10 provide strong evidence of absence.” (Page 21)

Behaviour statistics: explain how multiple comparisons were accounted for.

We used a 2 (time: pre-training, post-training) x 2 (group: ASD, TD) repeated measures ANOVA model to examine behavioral performance across time and group for each task and performed planned two-sample *t*-tests to assess group difference in performance. The Results section has been revised to clearly present our analysis approach (Page 21):

“For both of verification and production tasks, a 2x2 repeated measures ANOVA with time (pre-training, post-training) as a within-subject factor and group (ASD, TD) as a between-subject factor was performed. To further assess group differences in performance for each task, we conducted planned two-sample *t*-tests. ” (Page 21)

Intro paragraph: ASD and intellectual disability are contrasted as two separate things, yet often co-occur. Maybe moderate the language a little to reflect that.

Thank you for this suggestion. In the revised manuscript, we have modified this statement in the Introduction section to address comorbid conditions (Page 3):

“Relative to individuals with other neurodevelopment disorders, many individuals with ASD achieve lower levels of post-secondary education, employment, and independent living (Newman et al., 2011; Troyb et al., 2014).” (Page 3)

The acronym BF needs to be explained and contextualized.Intro to methods: be clearer about how the initial n=116 was reduced to 35 ASD and 28 TD.

We thank the reviewer for this comment. We have now included a more detailed description of participant inclusion procedures in the revised Methods and Figure 1 —figure supplement 1 (Page 16):

“All children completed neuropsychological assessments and their parents or legal guardians completed demographics, medical history, screening for MRI scan safety, and questionnaires regarding their child. In addition, children with ASD completed diagnostic assessments to confirm their autism diagnosis. Children’s eligibility to participate in the training study was determined by neuropsychological assessment scores (FSIQ > 80), ability to perform MRI protocol procedures, and autism diagnostic criteria. Twenty-one children did not meet the inclusion criteria and were not eligible to participate in the training study. Among eligible participants, 23 children discontinued due to a lack of interest or difficulties in coordinating their schedules. Eight children were unable to continue with the study due to challenges in following instructions during the sessions. Finally, one child was excluded due to task administration error. A total of 35 children with ASD and 28 TD children who completed training and pre/post behavioral and brain imaging sessions were included in the current study. A detailed visual depiction of participant inclusion procedures is provided in Figure 1 —figure supplement 1.” (Page 16)

Throughout the text, especially in the discussion: make it clearer whether they expect a unitary autism learning paradigm, or subsets of children with different learning strategies. That is, explore the role of the wide heterogeneity of the condition on learning. Also some discussion of whether they would expect this to be particular to autism or generalize to related neurodevelopmental disorders. (I realize that there is some discussion about IS as the mediating factor to explain heterogeneity, but I believe that there could be further emphasis and discussion throughout the text)

We thank the reviewer for raising this important point and allowing us to provide further clarification in the revised manuscript. While our central hypothesis is that learning mechanisms in ASD would be different from TD children, we agree that heterogeneity in response to intervention may be an important feature that differentiates children with ASD from their TD peers. In other words, there may not be a single mechanism that explains atypical learning in children with ASD, but rather affected children may learn differently from their TD peers in diverse ways. We have made every effort to highlight these points in the revised manuscript, including the Discussion (Page 12):

“It is important to note that heterogeneity in cognitive abilities and clinical symptomatology is a hallmark of ASD. Previous studies have consistently demonstrated that children with ASD exhibit a broad range of both cognitive strengths and challenges (Chen et al., 2019; Fountain et al., 2012; Gotham et al., 2009, 2012; Jones et al., 2009; Kjelgaard and Tager-Flusberg, 2001; Norbury and Nation, 2011; Wei et al., 2015), which likely stem from their atypical learning styles. Heterogeneity in response to intervention may be also one of important features that differentiate children with ASD from their TD peers. Previous studies have pointed to the possibility of different types of atypical learning in ASD. One possibility is that individuals with ASD rely more on memorizing specific examples rather than grasping general patterns (Qian and Lipkin, 2011). Another possibility relates to hyper-systemizing traits in ASD, which are associated with enhanced systematic and logical thinking and learning abilities (Baron-Cohen and Belmonte, 2005; Falter et al., 2008). Findings from our study suggest that children with ASD may benefit from consistent adherence to explicitly instructed rule-based learning, which aligns with the perspective that affected individuals rely on higher-than-average systematizing mechanisms that focus on identifying regularizes and rules. While our study provides evidence for the atypical learning style associated with hyper-systemizing traits in children with ASD, it is possible that there are multiple subtypes of atypical learning within this population. Future studies with a larger sample of children with ASD may help determine the presence of subtypes of distinct learning styles in affected children. Our findings may also hold relevance for other neurodevelopmental disorders, including ADHD which are also known to be characterized by substantial heterogeneity (Astle et al., 2022; Thapar et al., 2017). Further research addressing comorbidities and transdiagnostic overlap is required to determine whether atypical learning patterns observed in the current study are specific to autism or they extend to other neurodevelopmental disorders.” (Page 12)

Reviewer #2 (Recommendations for the authors):Overall I commend the authors for a nicely conducted intervention study, a clear and elegant series of analyses, and a compelling description of what is largely a null result (i.e., nearly identical training and NRP changes across groups). I think this is particularly timely, as there are a growing number of findings from our group and others that suggest many critical cognitive processes may be implemented differently, but remain relatively 'spared' in ASD relative to TD.To my reading, there are four primary weaknesses of this paper in its current form:1. The efficiency score does not seem to be calculated according to what is most often done in this literature. Typically, reaction time is divided by 1 minus the proportion of errors, and higher scores indicate LESS efficient performance (in the current paper, the key behavioral training efficiency was operationalized using accuracy divided by reaction time, with higher scores indicating MORE efficient performance). While these measures will be monotonically-related, based on some of my own back-of-the-envelope math as well as a series of formalizations in Bruyer and Brysbaert (2011; https://citeseerx.ist.psu.edu/document?repid=rep1andtype=pdfanddoi=1fcafd7a66caad684ae311d7468f4ca451c8dd0a), it seems likely that they may lead to distinct inferences when the proportion of errors are relatively large (e.g. >20%, as they are for several subjects here), or if there is not a high association between errors and response latency across subjects. I would appreciate it if the authors would either recalculate their efficiency measures so it is more in line with the formula that is commonly used in the literature [IES = RT / (1-prop error)]. Or, if the authors know there is precedent for the present formula that I am unaware of (quite possible!), some consideration of whether the current data meets the assumptions that must be met to support the use of the efficiency score in the first place (i.e., error rates largely <10-15%, between-subjects correlations between speed and accuracy, etc.) would be appreciated.

We thank the reviewer for their insightful comments. Considering familiarity with inverse efficiency score (IES) among cognitive researchers, we have replaced the efficiency scores with IES using its original formula, RT/(1-prop error), in the revised manuscript. The results using IES demonstrate similar findings as those reported using efficiency scores in the original manuscript, indicating that children with ASD exhibit comparable learning profiles across intervention sessions when compared to their typically developing (TD) peers. In the revised manuscript, we have replaced the efficiency score with IES on Pages 6, 19 and 21:

“On each day of training, performance on trained math problems was assessed using an inverse efficiency score (Bruyer and Brysbaert, 2011), measured as problem reaction time divided by accuracy. Higher scores on this measure indicated poorer performance. A 5x2 (Session x Group) repeated measures ANOVA showed a significant main effect of session (*F*(4, 212) = 41.82, *p* < 0.001, *η^2^_p_* = 0.44, BF >100), but no main effect of group, or session by group interaction (*F*s ≤ 0.09, *p*s ≥ 0.471, BFs < 0.33). A one-way repeated measures ANOVA indicated that both children with ASD and TD children showed significant changes in inverse efficiency scores across five days of training (*F*s ≥ 18.63, *p*s < 0.001, *η^2^_p_* ≥ 0.43, BFs ≥ 100) (Figure 2a and Supplementary File 3).” (Pages 6)

“Additionally, children’s learning rate across five training days, derived from a linear regression model, was comparable between children with ASD and TD children (*t*(53) = -0.86, *p* = 0.391, BF = 0.27). To further examine group differences in performance across training days, planned two-sample *t*-tests were performed for each training day. This analysis confirmed that inverse efficiency scores were not significantly different between groups on any training day (|*t*s| ≤ 0.84, *p*s ≥ 0.41, BFs < 0.46) (Supplementary File 4) (Pages 6)

“We computed a composite inverse efficiency score (IES) by dividing mean reaction time by accuracy (Bruyer and Brysbaert, 2011). The mean reaction time for each child on each training day was calculated using correctly solved trials.” (Pages 19)

“For computerized task administered on each day of training, we performed a 5x2 repeated measures ANOVA on inverse efficiency score with sessions (Day 1, 2, 3, 4, 5) as a within-subject factor and group (ASD, TD) as a between-subject factor. A one-way repeated measures ANOVA was performed on inverse efficiency score across sessions (Day 1, 2, 3, 4, 5) in each group.” (Pages 21)

Additionally, we conducted a careful examination of error rates and correlation between response times and error rates. Our analysis revealed that average error rates were lower than 0.41 and 0.23 in ASD and TD groups, respectively, and 90% of the error rates were not greater than 0.2. Importantly, even when we excluded six participants who had average error rates exceeding 0.2 on any of the five tutoring days, the results remained stable: a significant main effect of session (*F*(4, 188) = 64.58, *p* < 0.001, *η*^2^_p_ = 0.58), but no main effect of group (*F*(1, 47) = 0.12, *p* = 0.730) or session by group interaction (*F*(4, 188) = 0.71, *p* = 0.584) was observed. Finally, we calculated Pearson’s correlations between response times and error rates, which ranged from 0.59 to 0.77 across five tutoring days: Day 1: *r* = 0.77; Day 2: *r* = 0.70; Day 3: *r* = 0.62; Day 4: *r* = 0.62; Day 5: *r* = 0.59; all *p*s < 0.001. These values indicate overall low error rates and high correlations between response times and error rates, meeting the assumptions necessary for appropriate use of the inverse efficiency score (Bruyer and Brysbaert, 2011).

2. The current results comprise an ASD sample that has a mean IQ that is above the mean population IQ. This has limited generalizability to a significant proportion of individuals with ASD. This is common in the field, and I am sensitive to the logic behind IQ-matched samples, but some discussion of it as a limitation in the Discussion seems appropriate.

Thank you for bringing up this crucial point. We agree with the reviewer that it is unknown whether the findings will generalize to all children with ASD, including those with lower IQ. We now acknowledge this important point in the revised manuscript on Page 15:

“Second, our study focused on investigating atypical learning and associated neural mechanisms in children with ASD who had comparable cognitive abilities to our TD sample. Importantly, it remains unclear whether these mechanisms generalize to children with ASD with different levels of cognitive functioning, including children with lower IQ scores. It is hoped that future research addresses this question by including children with autism with a broader range of cognitive abilities.” (Page 15)

3. The study does not contain an adequate active control or sham condition to adequately account for non-intervention-related variance in behavior and BOLD data across pre-to-posttest. Therefore, it is impossible to know what the active ingredients of the intervention are. This was not a core aim of the work, so not a huge deal, but it would be good to, again, consider it as a potential limitation to be followed up in future studies.

Thank you for highlighting this important concern. We agree that inclusion of an appropriate active control intervention would be necessary to gain a better understanding of specific components of training that induce significant learning in children with ASD, in comparison to their TD peers. We have acknowledged this point as a limitation in the revised manuscript on Page 15-16, stating:

“Third, further studies incorporating an appropriate active control intervention, in both children with ASD and TD groups, are needed to gain a better understanding of specific components of training that induce significant learning in children with ASD.” (Page 16)

4. The authors argue that their behavioral and neural results may reflect the use of different dominant strategies to perform problem-solving tasks in ASD relative to TD. However, at this stage the dominant strategy models appear to be largely independent of the behavioral performance, NRP, and clinical inflexibility variables, making it hard to fully interpret them relative to the other data in the paper. The authors might want to conduct additional analyses to support this claim, for example, testing whether the strategic differences played any functional role in shaping the behavioral and neural changes from the pre-post test within or between groups. For example: does the degree to which individuals with ASD stuck to a rule-based strategy modulate the associations observed in Figures4 or 5? Is this similar or different to the impact of strategy on brain-behavior relationships in TD? etc.

Thank you for this helpful suggestion. We agree it was an open question as to whether there may be an association between persistent use of rule-based strategy and atypical learning and brain plasticity in children with ASD. To address this possibility, we performed additional brain-behavior (1) moderation and (2) correlation analyses, as described below.

(1) First, in the brain-behavior moderation analysis, we examined whether the observed atypical relation between neural representational plasticity (NRP) and learning gain in children with ASD could be explained by *strategy transition status* – i.e., whether (a) children continued to primarily use the same (rule-based) strategy or (b) shifted to a different (memory-based) strategy following training. Specifically, we performed a moderation analysis in each group (ASD or TD) with learning gains as the dependent variable, NRP as the independent variable, and the strategy transition status as the moderator. For the strategy transition status, we assigned each child a status of either "transition" or "non-transition" based on whether or not their dominant strategy changed from rule-based to memory-based strategy between pre- and post-training sessions. Moderation analysis was performed for each region that showed significant group differences in brain-behavior (NRP-learning gain) relation, which included the bilateral MTL, right IPS, right lateral occipital cortex, right frontal eye field (FEF), and right middle frontal gyrus (MFG) (see also Figure 4 and Supplementary File 6).

This analysis revealed that in the ASD group, the strategy transition status moderated the *negative* relationship between NRP and learning gains in the right FEF (b = -0.48, se = 0.19, t = -2.58, p = 0.022, model p = 0.005; Figure 5 —figure supplement 1), with a greater negative association for individuals with *non-transition* status. This finding suggests that the observed atypical brain-behavior relation in children with ASD was likely driven by individuals whose dominant strategy remained the same (rule-based) from pre- to post-training session. No other ROI showed a significant moderation effect in children with ASD (*p*s > 0.084), which indicates specificity of these findings in a prefrontal cortical region implicated in visuospatial attention (Ronconi et al., 2014). Furthermore, in the TD group, we did not observe significant moderation effect of strategy transition status in any region (*p*s > 0.109), likely due to the fact that few children had non-transition status (6 out of 28 children) in this group. Together, these findings suggest that persistent use of rule-based strategy enhanced atypical patterns of brain-behavior relation in children with ASD. However, these observations were based on relatively small samples of "transition" and "non-transition" subgroups, and further studies are required to validate these findings in a larger sample.

(2) Second, in brain-behavior correlation analysis, we examined direct links between NRP and the degree of training-induced changes in the use of rule-based strategy. Here, we quantified rule-based strategy persistence as the extent to which children with ASD continued to rely on rule-based problem-solving strategy across pre- and post-training sessions, by subtracting the proportion of rule-based strategy use across all trained problems at pre-training from that at post-training. Values ranged from -1 to 0, with -1 indicating a shift from rule-based to memory-based problem-solving strategy for all of the trained problems and 0 indicating no change in strategy use for any of the trained problems following training. Thus, larger values (i.e., those closer to 0) represented persistence using a rule-based strategy across sessions. We performed Pearson’s correlation between rule-based strategy persistence and NRP for each ROI in children with ASD.

This analysis revealed a significant negative correlation between rule-based strategy persistence and NRP in the right MFG (r = -0.57, p = 0.015; Figure 4 —figure supplement 4), which indicates that children with ASD who persisted more on using rule-based strategy across trained problems were those who showed more reduced NRP, or greater stability in neural representation across pre- and post-training sessions. No other ROI showed a significant correlation (*p*s > 0.183), suggesting specificity of these findings in a prefrontal cortical region implicated in cognitive flexibility (Sami et al., 2023). In the TD group, we did not find a significant correlation between rule-based strategy persistence and NRP for any region (*p*s > 0.161). This result may have been influenced by the distribution of rule-based strategy persistence in TD children, many of whom did not exhibit persistence using a rule-based strategy. Together, these results suggest a direct link between increased tendency to persist on using rule-based strategies and higher level of stability in neural representation in children with ASD.

In the revised Discussion section (Pages 13-14), we now include these additional analyses (Figure 4 —figure supplement 4 and Figure 5 —figure supplement 1) which provide a new level of detail regarding our findings of atypical learning and brain plasticity in children with ASD:

“As discussed above, these findings suggest that, unlike TD children, children with ASD may develop efficient problem-solving procedures through continued repetition and execution of rule-based strategies. While the flexible use of different problem-solving strategies or transitions between strategies is advantageous for TD children, children with ASD demonstrate greater improvement using a less flexible approach.

Consistent with these hypotheses, we found a significant negative association between persistent use of rule-based strategy and NRP in the right middle frontal gyrus in children with ASD (Figure 4 —figure supplement 4). Furthermore, moderation analyses (Figure 5 —figure supplement 1) demonstrated that the association between lower NRP (i.e., increased stability in neural representation) and greater learning gains in the right FEF in children with ASD was driven by individuals who persistently used rule-based strategies across training sessions. In contrast, we did not observe such associations in TD children. These findings suggest that persistent use of rule-based strategy enhanced atypical patterns of brain-behavior relation in children with ASD. It is noteworthy that these findings were observed in prefrontal cortical regions implicated in visuospatial attention (Ronconi et al., 2014) and cognitive flexibility (Sami et al., 2023). Together, results from the current study highlight atypical patterns of problem-solving strategy use and brain plasticity that support preserved learning and identify neural sources of variability in cognitive skill acquisition in children with ASD in the context of educationally meaningful training.” (Pages 13)

Reviewer #3 (Recommendations for the authors):The discussion of savants and exceptional abilities in the introduction was interesting and it may be beneficial to include such information in the analyses or as part of the participant information section. For instance, were there any autistic individuals in the study with exceptional skills and did they then also rely on the same memory-based strategies as the average? It may therefore be beneficial to include an individual approach or description.

We appreciate the reviewer's insightful comment regarding savants and exceptional abilities in autism. While this is an intriguing topic for future investigation, our study was not specifically designed to assess exceptional abilities in this population. We note below our observation from limited assessments of children (IQ) and their parents (survey).

First, standardized assessment of their IQ indicated that children with ASD in the current study had a comparable range of scores as TD children. Furthermore, IQ was not significantly associated with learning gains (*r* = -0.10, *p* = 0.597) or change in strategy use (two-sample t-test, *t*(31) = 1.27, *p* = 0.212), which indicates that findings were not likely driven by children with autism with exceptionally high IQ. However, it is worth noting that assessments of IQ were not intended to assess exceptional abilities, and it is possible children with ASD had exceptional skills in areas that were not assessed.

Second, as part of a comprehensive parent survey of their children, the *Special Skill in Autism Questionnaire* was administered to participating children’s parents. Parents’ participation in the survey was voluntary, and 60% of parents responded to this survey, including the parents of 21 participants in the final ASD sample. Of these children with ASD, 28% indicated good math or computational/algorithmic skills, 15% reported musical proficiency, 10% mentioned strong reading abilities, 10% reported expertise in science or computer skills, 15% reported excellent memory skills, 5% mentioned visual recognition skills, 5% reported engagement in daily games, and 15% did not report any outstanding skills for their child.

Although results from the parent survey indicate potential presence of exceptional abilities within our sample of ASD, standardized assessments measured across a wide range of domains may be necessary to determine whether their self-reported skills are exceptional compared to expected range of skills at this stage of development. Due to limited available data, we were not able to address potential relationships between exceptional skills and behavioral or brain measures examined in the current study. We appreciate the reviewer’s thoughtful feedback and believe that future research focusing on this topic may help determine whether children with autism with exceptional abilities have distinct profiles of learning or shift in strategy use across training.

The investigation of insistence on sameness as a moderator of brain plasticity and learning was a valuable analysis. Did the authors consider a brain region as a potential moderator or mediator such as the insula or other behaviors such as cognitive flexibility/shifting?

The goal of moderation analysis was to identify factors that modulate the degree of atypical learning-related brain plasticity in children with ASD. Thus, we focused on brain regions identified from whole brain analysis of group by learning gain interaction (i.e., regions that showed significant group differences in brain-behavior relation; see also Figure 4 and Supplementary File 6). These regions included the MTL and IPS, which have been consistently implicated in numerical problem solving and learning (Butterworth and Walsh, 2011; Menon and Chang, 2021). While we focused on the MTL and IPS regions due to their strong associations with mathematical learning, we additionally confirmed that findings from our moderation analysis are specific to MTL and IPS region by performing control analyses. Specifically, we found no significant moderation effect for either model that included NRP of a visual region (V1) or the whole brain. Furthermore, as suggested by the reviewer, we examined whether NRP in the insula could moderate atypical relation between learning gain and NRP in the MTL or IPS in children with ASD. There was no significant moderation effect of NRP in either the left or right insula on the relation between learning gains and NRP in the MTL (*p*s >.07) or IPS (*p*s >.27). Future studies may systematically examine whether and how NRP of different regions interact and contribute to atypical learning in children with ASD.

We acknowledge the possibility that additional brain regions may also be engaged in learning-related brain plasticity in children with ASD, and that the relation between RRIB and learning could be moderated or mediated by brain plasticity. However, our analysis was focused on addressing whether RRIB moderates *atypical* learning and brain plasticity in these children. Therefore, we did not pursue exploratory, whole-brain analysis or additional paths of moderation or mediation analysis to avoid potential concerns regarding enhanced false positive rates.

We did not include a measure of cognitive flexibility or shifting in our study. However, previous studies have provided evidence in support of cognitive inflexibility in ASD (Crawley et al., 2020; Geurts et al., 2009; Uddin, 2021) which has been closely linked to insistence on sameness (IS) (Lam et al., 2008; Miller et al., 2015; Supekar et al., 2021a). We agree that future studies should assess this relation more directly by administering a comprehensive assay of cognitive flexibility measures in children with ASD. In the revised manuscript, we have included this point as a consideration for future studies (Page 16):

“Fourth, identifying brain mechanisms underlying cognitive inflexibility and its relation to cognitive skill acquisition in children with ASD is an important area for future research.” (Page 16)

Additionally, was a total RRIB score used in any of the analyses instead of individual RRIBs? Or perhaps elaborate on why different forms of RRIBs were not significant.

We examined total RRIB scores derived from ADI-R and found that the total score did not significantly moderate the atypical brain-behavior relation (b = -0.29, se = 0.20, t = -1.46, p = 0.167; model p = 0.062). We further examined three RRIB sub-scores derived from the ADI-R, including IS, circumscribed interests, and repetitive motor behavior. Based on prior research that IS is closely linked to the construct of cognitive flexibility (Lam et al., 2008; Miller et al., 2015; Supekar et al., 2021a), we reasoned that IS would be most closely associated with atypical learning among these RRIB components as cognitive flexibility may influence individuals’ learning styles (e.g., more flexible/adaptive vs. less flexible/adaptive learning). The specificity of IS findings in our study, and lack of significant effects related to circumscribed interests or repetitive motor behavior, provide additional support for dissociation of multiple components of RRIBs. In the revised manuscript, we clarify the rationale for examining multiple components of RRIBs, and specifically insistence on sameness, in Introduction and Discussion sections on Page 5-6 and 14:

“Although behavioral studies have proposed a connection between RRIB and flexible behavior during probabilistic reversal learning tasks (Crawley et al., 2020; D'Cruz et al., 2013), the neural mechanisms linking RRIB to learning have not been explored. To determine whether RRIB contributes to atypical mechanisms of learning in children with ASD, we evaluated the hypothesis that RRIB symptoms would influence the relation between brain plasticity and learning in these children. We were particularly interested in the contribution of insistence on sameness (IS), a core phenotypic feature of ASD related to cognitive and behavioral inflexibility and resistance to changes in routine (Lam et al., 2008; Supekar et al., 2021b). We hypothesized that, among the three RRIB elements, IS would be most closely associated with atypical learning patterns, as cognitive flexibility – which can vary from highly flexible and adaptive to less flexible and adaptive – has the potential to significantly shape an individual's learning style.” (Page 5-6)

“Furthermore, the moderating role of IS on the relation between learning and brain plasticity was not observed with respect to the two other RRIB components, circumscribed interests and repetitive motor behaviors. A possible explanation for this finding is that the IS construct has a clear conceptual overlap with cognitive and behavioral inflexibility and has been specifically linked to cognitive set shifting deficits (Miller et al., 2015). Our findings suggest that higher levels of IS in ASD may impair mental flexibility including the ability to make strategy adjustments as a means of improving performance during learning. Moreover, previous studies have shown that IS is distinct from sensory-motor contributions to RRIB (Miller et al., 2015; Mooney et al., 2009). Our findings converge on these reports and provide evidence for the link between neural plasticity, learning, and the IS component of RRIB. Extending previous proposals of heterogeneity in learning in ASD (Church et al., 2015; Mercado and Church, 2016), our findings reveal IS as a key source of such heterogeneity and are consistent with a previous observation of less adaptable patterns of brain activity during learning that are associated with ASD symptoms (Schipul and Just, 2016).” (Page 14)

Further, why were RRIBs used rather than the score from the ADOS RRB?

This choice was based on the hypotheses described above, and prior research demonstrating a robust factor structure of the ADI-R RRIB comprising cognitive and motoric factors (Lam et al., 2008; Miller et al., 2015; Mooney et al., 2009) as well as our prior research linking the IS subscores of RRIBs from the ADI-R to cognitive inflexibility in autism (Supekar et al., 2021a). In contrast, the ADOS does not separately assess cognitive and motoric components of RRIB.

The finding that learning is supported by greater stability of neural representations in autism could be further discussed.

Thank you for your feedback. In the revised manuscript, we have expanded the Discussion on Page 13-14 to discuss this further:

“Our finding that greater stability of neural representations is associated with greater learning in children with ASD has important implications in the context of persistence of rule-based strategies. We hypothesize that learning in children with ASD may be supported by stability of problem-solving strategies and neural representations across training sessions, which is broadly consistent with the notion that typical behavioral performance in ASD is achieved by atypical neural mechanisms (Church et al., 2015; Dovgopoly and Mercado, 2013; Livingston and Happe, 2017). As discussed above, these findings suggest that, unlike TD children, children with ASD may develop efficient problem-solving procedures through continued repetition and execution of rule-based strategies. While the flexible use of different problem-solving strategies or transitions between strategies is advantageous for TD children, children with ASD may show greater improvement using a less flexible approach.

Consistent with these hypotheses, we found a significant negative association between persistent use of rule-based strategy and NRP in the right middle frontal gyrus in children with ASD (Figure 4 —figure supplement 4). Furthermore, moderation analyses (Figure 5 —figure supplement 1) demonstrated that the association between lower NRP (i.e., increased stability in neural representation) and greater learning gains in the right FEF in children with ASD was driven by individuals who persistently used rule-based strategies across training sessions. In contrast, we did not observe such associations in TD children. These findings suggest that persistent use of rule-based strategy enhanced atypical patterns of brain-behavior relation in children with ASD. It is noteworthy that these findings were observed in prefrontal cortical regions implicated in visuospatial attention (Ronconi et al., 2014) and cognitive flexibility (Sami et al., 2023). Together, results from the current study highlight atypical patterns of problem-solving strategy use and brain plasticity that support preserved learning and identify neural sources of variability in cognitive skill acquisition in children with ASD in the context of educationally meaningful training. ”(Page 13-14)

"Data from 13 children with ASD and 4 TD children were excluded in the following analysis due to excessive head motion (translation or rotation above 10mm or 10{degree sign} in any direction or mean frame-wise head motion above 0.5mm) in more than half of runs. Data from runs with excessive head motion were also excluded for the following analysis." More information about the head motion profiles for subjects would be useful, including if there were significant differences in head motion between the groups. And this information may be included in supplementary File 1.

Thank you for this suggestion. We now include additional information about children’s head movement in Supplementary File 1. There were no significant differences in transitional (x, y, z) and rotational (pitch, roll, yaw) head movement parameters between the ASD and TD groups in both pre-training (all |*t*s*|* < 1.33, *p*s > 0.191) and post-training (all |*t*s*|* < 1.84, *p*s > 0.073) sessions. For mean frame-wise displacement, there was no significant difference between groups in the pre-training session (*t(43)* = 1.41, *p* = 0.165), but in the post-training session, children with ASD showed higher values (*t(43)* = 2.22, *p* = 0.032). For maximum frame-wise displacement, there were no significant group differences in both pre-training (*t(43)* = 1.16, *p* = 0.254) and post-training (*t(43)* = 1.41, *p* = 0.165) sessions. Head motion parameters were included as covariates of no interest in fMRI general linear model of first-level statistical analysis for task-related brain activation.

To account for potential influences of mean frame-wise displacement between groups, we performed additional brain-behavior association analysis with mean frame-wise displacement averaged across time points as a covariate. This analysis replicated our main findings of atypical relations between neural representational plasticity and learning gains in children with ASD. Specifically, the dissociable brain-behavior relationships between children with ASD and TD children were also observed in the MTL, IPS, lateral occipital cortex, frontal eye field, and middle frontal gyrus (Author response image 1):

**Author response image 1. sa2fig1:** The Results of the brain-behavior relation after mean frame-wise displacement regression. Whole-brain multivariate neural representational pattern analysis revealed a significant group by learning gain interaction in the right medial temporal lobe (MTL), right intraparietal sulcus (IPS), right lateral occipital cortex (LOC), right frontal eye field (FEF), and right middle frontal gyrus (MFG). NRP in each region identified from the interaction between group and learning gains in whole-brain analysis was extracted from 6-mm spheres centered at peaks for visualization of results. Shaded areas indicate 95% confidence intervals.

A further description of the RRIB measures, such as what each of them captures, could be included in the method or supplementary materials beyond the 2021 reference in the methods section.

In the revised manuscript, we have included a description of RRIB subscore measures in the Methods section (Page 24):

“Circumscribed interests and insistence on sameness subscores capture children’s adherence to specific interests and routines and are considered to be cognitive components of RRIB. In contrast, the repetitive motor component encompasses actions such as hand flapping, rocking, and head banging, behaviors which are considered motoric in origin (Lewis et al., 2007; Turner, 1999).” (Page 24)

Figure 1: More description of the tasks could be included in the figure legend. Further, details on what fMRI scan was used (pre v post-training) should also be included.

We appreciate the reviewer's detailed feedback. The updated figure legend includes the following information about the tasks and fMRI scan:

“In the math verification task, each trial began with a 500ms fixation cross, followed by a 6-second presentation of a math problem. Participants considered the problem and then indicated if a subsequent solution probe matched their answer. In the math production task, children were required to solve each addition problem by verbalizing their answers. After solving each problem, children’s problem-solving strategies were assessed, in which they were asked to describe how they solved the problem. Each task fMRI run lasted 4 minutes and 50 seconds.” (Page 37)

Figure 2: It appears that there is a near-equal split between rule-based and memory-based strategies used in the ASD group. This could be further clarified in the results or perhaps emphasized – regarding greater reliance on rule-based methods and the untrained group (Figure 2d).

Thank you for bringing this to our attention. We would like to provide further clarification regarding the reliance on rule-based problem-solving strategy in children with ASD within the context of our study. The greater reliance on rule-based strategy in children with ASD was observed in comparison to TD children. Specifically, compared to their typically developing (TD) peers, children with ASD were relatively less likely to rely on memory-based strategy and were more likely to rely on rule-based strategy following the same training protocol which was designed to promote memory-based problem-solving strategy. While it is plausible that some children with ASD may have changed from relying on rule-based strategy to increased use of memory-based strategy as a result of training, though at a reduced rate than TD children, the χ2 statistic testing this hypothesis was not significant. To clarify these points, we have included additional information in the Results section of the revised manuscript (Page 8), where we state:

“These results demonstrate that despite similar improvements in post-training performance between children with ASD and TD children, the problem-solving strategies employed by these two groups diverged significantly. Specifically, even with a training protocol designed to encourage memory-based problem-solving strategies, children with ASD showed a greater tendency to rely on rule-based learning for trained problems, as opposed to the memory-based learning more frequently employed by TD children. Additionally, children with ASD exhibited less variation in strategy use between trained and untrained problems due to training, in comparison to their TD peers.” (Page 8)

Furthermore, in the figure legend of Figure 2 (Page 38), we have clarified this point by stating:

“While most TD children used the memory-based strategy most frequently following training, nearly half of the children with ASD used rule-based strategies most frequently for trained problems.” (Page 38)

Additionally, we have highlighted the distinction in strategy use between trained and untrained problems in children with ASD compared to TD children in the figure legend:

“After training, the distribution of strategy use was not significantly distinguishable between trained and untrained problems in children with ASD, whereas TD children reported significantly greater use of memory-retrieval strategy for trained problems compared to untrained problems.” (Page 38)

Figure 4: Unfortunately the text is too small and blurry to really assess this figure. However, it appears that the blue line (ASD group) is declining. However, in the manuscript, it is stated that the neural changes are "stable". Please clarify/emphasize the use of the word "stable" in relation to the plots.

We apologize for any confusion caused by the low resolution figure in our original submission. We have now replaced this and all other figures with high resolution figures with larger text in our revised manuscript.

We would like to clarify our interpretation of “stable neural representations” based on the degree of neural representational plasticity (NRP) measure as well as our findings of group differences in NRP (Figure 3) and the relation of NRP and learning gain (Figure 4).

NRP was a brain-based distance measure to quantify spatial correlation in brain activity patterns during math problem solving before and after training. NRP scores were normalized values of correlation between pre- and post-training multiplied by -1. Higher NRP scores indicated a greater degree of training-induced brain plasticity, while lower NRP scores indicated a lower degree of plasticity in neural representation between pre- and post-training, reflecting greater stability of neural representations over time.

As shown in Figure 3, overall levels of NRP were comparable between children with ASD and TD children, which indicated similar levels of neural representational plasticity (or stability) between the groups without the consideration of individual differences in learning. However, remarkable differences between the groups were observed with regards to the relation between NRP and learning gains. Specifically, the scatter plots in Figure 4 illustrate distinct patterns of brain-behavior (i.e., NRP and learning gains) relationships between the two groups of children, with increasing slope indicating a positive relation and decreasing slope indicating a negative relation. In TD children, shown in the pink line (depicting increasing slope), higher learning gains were associated with greater NRP. In children with ASD, shown in the blue line (depicting decreasing slope), higher learning gains were associated with lower NRP, which was considered to reflect more stable neural representation as described above.

We have now included clarifications in the figure legend of Figure 4 as follows (Page 40):

“NRP scores were normalized values of correlation between pre- and post-training multiplied by -1. Larger NRP scores indicated a greater degree of training-related brain plasticity, while smaller NRP scores indicated a lower degree of plasticity in neural representation between pre- and post-training, reflecting increased stability of neural representations over time. In typically developing (TD) children, shown in the pink line, greater learning gains were associated with greater NRP. In children with autism spectrum disorder (ASD), shown in the blue line, greater learning gains were associated with lower NRP, indicative of more stable neural representations.” (Page 40)

References

Astle, D. E., Holmes, J., Kievit, R., and Gathercole, S. E. (2022). Annual Research Review: The transdiagnostic revolution in neurodevelopmental disorders. *J Child Psychol Psychiatry, 63*(4), 397-417. https://doi.org/10.1111/jcpp.13481

Baron-Cohen, S., and Belmonte, M. K. (2005). Autism: a window onto the development of the social and the analytic brain. *Annu Rev Neurosci, 28*, 109-126. https://doi.org/10.1146/annurev.neuro.27.070203.144137

Bruyer, R., and Brysbaert, M. (2011). Combining speed and accuracy in cognitive psychology: Is the inverse efficiency score (IES) a better dependent variable than the mean reaction time (RT) and the percentage of errors (PE)? *Psychologica Belgica, 51*(1), 5-13. http://hdl.handle.net/2078.1/163356

Butterworth, B., and Walsh, V. (2011). Neural basis of mathematical cognition. *Curr Biol, 21*(16), R618-621. https://doi.org/10.1016/j.cub.2011.07.005

Chen, L., Abrams, D. A., Rosenberg-Lee, M., Iuculano, T., Wakeman, H. N., Prathap, S., Chen, T., and Menon, V. (2019). Quantitative analysis of heterogeneity in academic achievement of children with autism. *Clin Psychol Sci, 7*(2), 362-380. https://doi.org/10.1177/2167702618809353

Church, B. A., Rice, C. L., Dovgopoly, A., Lopata, C. J., Thomeer, M. L., Nelson, A., and Mercado, E., 3rd. (2015). Learning, plasticity, and atypical generalization in children with autism. *Psychon Bull Rev, 22*(5), 1342-1348. https://doi.org/10.3758/s13423-014-0797-9

Crawley, D., Zhang, L., Jones, E. J. H., Ahmad, J., Oakley, B., San Jose Caceres, A., Charman, T., Buitelaar, J. K., Murphy, D. G. M., Chatham, C., den Ouden, H., Loth, E., and group, E.-A. L. (2020). Modeling flexible behavior in childhood to adulthood shows age-dependent learning mechanisms and less optimal learning in autism in each age group. *PLoS Biol, 18*(10), e3000908. https://doi.org/10.1371/journal.pbio.3000908

D'Cruz, A. M., Ragozzino, M. E., Mosconi, M. W., Shrestha, S., Cook, E. H., and Sweeney, J. A. (2013). Reduced behavioral flexibility in autism spectrum disorders. *Neuropsychology, 27*(2), 152-160. https://doi.org/10.1037/a0031721

Dovgopoly, A., and Mercado, E., 3rd. (2013). A connectionist model of category learning by individuals with high-functioning autism spectrum disorder. *Cogn Affect Behav Neurosci, 13*(2), 371-389. https://doi.org/10.3758/s13415-012-0148-0

Falter, C. M., Plaisted, K. C., and Davis, G. (2008). Visuo-spatial processing in autism--testing the predictions of extreme male brain theory. *J Autism Dev Disord, 38*(3), 507-515. https://doi.org/10.1007/s10803-007-0419-8

Fountain, C., Winter, A. S., and Bearman, P. S. (2012). Six developmental trajectories characterize children with autism. *Pediatrics, 129*(5), e1112-1120. https://doi.org/10.1542/peds.2011-1601

Geurts, H. M., Corbett, B., and Solomon, M. (2009). The paradox of cognitive flexibility in autism. *Trends Cogn Sci, 13*(2), 74-82. https://doi.org/10.1016/j.tics.2008.11.006

Gotham, K., Pickles, A., and Lord, C. (2009). Standardizing ADOS scores for a measure of severity in autism spectrum disorders. *J Autism Dev Disord, 39*(5), 693-705. https://doi.org/10.1007/s10803-008-0674-3

Gotham, K., Pickles, A., and Lord, C. (2012). Trajectories of autism severity in children using standardized ADOS scores. *Pediatrics, 130*(5), e1278-1284. https://doi.org/10.1542/peds.2011-3668

Jones, C., Happe, F., Golden, H., Marsden, A. J., Tregay, J., Simonoff, E., Pickles, A., Baird, G., and Charman, T. (2009). Reading and arithmetic in adolescents with autism spectrum disorders: peaks and dips in attainment. *Neuropsychology, 23*(6), 718-728. https://doi.org/10.1037/a0016360

Kjelgaard, M. M., and Tager-Flusberg, H. (2001). An Investigation of Language Impairment in Autism: Implications for Genetic Subgroups. *Lang Cogn Process, 16*(2-3), 287-308. https://doi.org/10.1080/01690960042000058

Lam, K. S., Bodfish, J. W., and Piven, J. (2008). Evidence for three subtypes of repetitive behavior in autism that differ in familiality and association with other symptoms. *J Child Psychol Psychiatry, 49*(11), 1193-1200. https://doi.org/10.1111/j.1469-7610.2008.01944.x

Lewis, M. H., Tanimura, Y., Lee, L. W., and Bodfish, J. W. (2007). Animal models of restricted repetitive behavior in autism. *Behav Brain Res, 176*(1), 66-74. https://doi.org/10.1016/j.bbr.2006.08.023

Livingston, L. A., and Happe, F. (2017). Conceptualising compensation in neurodevelopmental disorders: Reflections from autism spectrum disorder. *Neurosci Biobehav Rev, 80*, 729-742. https://doi.org/10.1016/j.neubiorev.2017.06.005

Menon, V., and Chang, H. (2021). Emerging neurodevelopmental perspectives on mathematical learning. *Dev Rev, 60*. https://doi.org/10.1016/j.dr.2021.100964

Mercado, E., 3rd, and Church, B. A. (2016). Brief Report: Simulations Suggest Heterogeneous Category Learning and Generalization in Children with Autism is a Result of Idiosyncratic Perceptual Transformations. *J Autism Dev Disord, 46*(8), 2806-2812. https://doi.org/10.1007/s10803-016-2815-4

Miller, H. L., Ragozzino, M. E., Cook, E. H., Sweeney, J. A., and Mosconi, M. W. (2015). Cognitive set shifting deficits and their relationship to repetitive behaviors in autism spectrum disorder. *J Autism Dev Disord, 45*(3), 805-815. https://doi.org/10.1007/s10803-014-2244-1

Mooney, E. L., Gray, K. M., Tonge, B. J., Sweeney, D. J., and Taffe, J. R. (2009). Factor analytic study of repetitive behaviours in young children with Pervasive Developmental Disorders. *J Autism Dev Disord, 39*(5), 765-774. https://doi.org/10.1007/s10803-008-0680-5

Newman, L., Wagner, M., Knokey, A.-M., Marder, C., Nagle, K., Shaver, D., and Wei, X. (2011). The Post-High School Outcomes of Young Adults with Disabilities up to 8 Years after High School: A Report from the National Longitudinal Transition Study-2 (NLTS2). NCSER 2011-3005. *National Center for Special Education Research*.

Norbury, C., and Nation, K. (2011). Understanding Variability in Reading Comprehension in Adolescents With Autism Spectrum Disorders: Interactions With Language Status and Decoding Skill. *Scientific Studies of Reading, 15*(3), 191-210. https://doi.org/10.1080/10888431003623553

Qian, N., and Lipkin, R. M. (2011). A learning-style theory for understanding autistic behaviors. *Front Hum Neurosci, 5*, 77. https://doi.org/10.3389/fnhum.2011.00077

Ronconi, L., Basso, D., Gori, S., and Facoetti, A. (2014). TMS on right frontal eye fields induces an inflexible focus of attention. *Cerebral Cortex, 24*(2), 396-402. https://doi.org/10.1093/cercor/bhs319

Rouder, J. N., Morey, R. D., Speckman, P. L., and Province, J. M. (2012). Default Bayes factors for ANOVA designs. *Journal of mathematical psychology, 56*(5), 356-374. https://doi.org/10.1016/j.jmp.2012.08.001

Rouder, J. N., Speckman, P. L., Sun, D., Morey, R. D., and Iverson, G. (2009). Bayesian t tests for accepting and rejecting the null hypothesis. *Psychon Bull Rev, 16*(2), 225-237. https://doi.org/10.3758/PBR.16.2.225

Sami, H., Tei, S., Takahashi, H., and Fujino, J. (2023). Association of cognitive flexibility with neural activation during the theory of mind processing. *Behav Brain Res, 443*, 114332. https://doi.org/10.1016/j.bbr.2023.114332

Schipul, S. E., and Just, M. A. (2016). Diminished neural adaptation during implicit learning in autism. *Neuroimage, 125*, 332-341. https://doi.org/10.1016/j.neuroimage.2015.10.039

Supekar K, Chang H, Mistry PK, Iuculano T, Menon V. 2021a. Neurocognitive modeling of latent memory processes reveals reorganization of hippocampal-cortical circuits underlying learning and efficient strategies. *Communications Biology*
**4**:405. 10.1038/s42003-021-01872-1, 33767350

Supekar, K., Ryali, S., Mistry, P. K., and Menon, V. (2021b). Aberrant dynamics of cognitive control and motor circuits predict distinct restricted and repetitive behaviors in children with autism. *Nat Commun, 12*(1), 3537. https://doi.org/10.1038/s41467-021-23822-5

Thapar, A., Cooper, M., and Rutter, M. (2017). Neurodevelopmental disorders. *Lancet Psychiatry, 4*(4), 339-346. https://doi.org/10.1016/S2215-0366(16)30376-5

Troyb, E., Orinstein, A., Tyson, K., Helt, M., Eigsti, I. M., Stevens, M., and Fein, D. (2014). Academic abilities in children and adolescents with a history of autism spectrum disorders who have achieved optimal outcomes. *Autism, 18*(3), 233-243. https://doi.org/10.1177/1362361312473519

Turner, M. (1999). Annotation: Repetitive behaviour in autism: a review of psychological research. *J Child Psychol Psychiatry, 40*(6), 839-849. https://www.ncbi.nlm.nih.gov/pubmed/10509879

Uddin, L. Q. (2021). Brain Mechanisms Supporting Flexible Cognition and Behavior in Adolescents With Autism Spectrum Disorder. *Biol Psychiatry, 89*(2), 172-183. https://doi.org/10.1016/j.biopsych.2020.05.010

Wagenmakers, E. J., Lodewyckx, T., Kuriyal, H., and Grasman, R. (2010). Bayesian hypothesis testing for psychologists: a tutorial on the Savage-Dickey method. *Cogn Psychol, 60*(3), 158-189. https://doi.org/10.1016/j.cogpsych.2009.12.001

Wei, X., Christiano, E. R., Yu, J. W., Wagner, M., and Spiker, D. (2015). Reading and math achievement profiles and longitudinal growth trajectories of children with an autism spectrum disorder. *Autism, 19*(2), 200-210. https://doi.org/10.1177/1362361313516549